# You Are What You Annotate:
# Towards Better Models through Annotator Representations

**Naihao Deng**🍑   **Xinliang Frederick Zhang**🍑   **Siyang Liu**🍑
**Winston Wu**🍋   **Lu Wang**🍑   **Rada Mihalcea**🍑

🍑University of Michigan – Ann Arbor, USA   🍋University of Hawai'i at Hilo, USA
{dnaihao, mihalcea}@umich.edu

## Abstract

Annotator disagreement is ubiquitous in natural language processing (NLP) tasks. There are multiple reasons for such disagreements, including the subjectivity of the task, difficult cases, unclear guidelines, and so on. Rather than simply aggregating labels to obtain data annotations, we instead try to directly model the diverse perspectives of the annotators, and explicitly account for annotators' idiosyncrasies in the modeling process by creating representations for each annotator (*annotator embeddings*) and also their annotations (*annotation embeddings*). In addition, we propose **TID-8**, **T**he **I**nherent **D**isagreement - **8** dataset, a benchmark that consists of eight existing language understanding datasets that have inherent annotator disagreement. We test our approach on TID-8 and show that our approach helps models learn significantly better from disagreements on six different datasets in TID-8 while increasing model size by fewer than 1% parameters. By capturing the unique tendencies and subjectivity of individual annotators through embeddings, our representations prime AI models to be inclusive of diverse viewpoints.

## 1 Introduction

Annotator disagreement is a common challenge in NLP (Leonardelli et al., 2021; Fornaciari et al., 2021). The conventional approach to reconciling such disagreements is to assume there is a single ground-truth label, and aggregate annotator labels on the same data instance (Paun and Simpson, 2021). However, disagreement among annotators can arise from various factors, including differences in interpretation, certain preferences (e.g. due to annotators' upbringing or ideology), difficult cases (e.g., due to uncertainty or ambiguity), or multiple plausible answers (Plank, 2022). It is problematic to simply treat disagreements as noise and reconcile the disagreements by aggregating different labels into a single one. To illustrate, consider the case of hate speech detection, where certain

| **Humor** |
| --- |
| *Text A:* Being crushed by large objects can be very depressing. |
| *Text B:* As you make your bed, so you will sleep on it. |
| ANN WHICH IS FUNNIER, X MEANS A TIE: A, A, B, X, X |

Table 1: An example where annotators disagree. Table 7 in Appendix B.1 shows more examples.

words or phrases might be harmful to specific ethnic groups (Kirk et al., 2022). For instance, terms that white annotators regard as innocuous might be offensive to black or Asian annotators due to cultural nuances and experiences that shape the subjective perceptions of hate speech. Adjudication over the annotation of hate speech assumes that there is a standard "correct" way people should feel towards these texts, which ignores under-represented groups whose opinions may not agree with the majority. Similarly, in humor detection, different people can have varying levels of amusement towards the same text (Ford et al., 2016; Jiang et al., 2019), making it difficult to reach a consensus on such a subjective task. Another example is natural language inference (NLI), where it has been shown that there are inherent disagreements in people's judgments that cannot be smoothed out by hiring more workers (Pavlick and Kwiatkowski, 2019). Aggregating labels in NLI tasks can disregard the reasoning and perspective of certain individuals, undermining their intellectual contributions.

To account for these disagreements, one approach is to directly learn from the data that has annotation disagreements (Uma et al., 2021), but representing this information inside the models is often not trivial. Instead, to leverage the diverse viewpoints brought by different annotators, we create representations for the annotators (*annotator embeddings*) and for their annotations (*annotation embeddings*), with learnable matrices associated with both of these embeddings (see Section 4). On downstream tasks, we forward the weighted embed-

dings together with the text embeddings to the classification model, which adjusts its prediction for each annotator. Intuitively, by modeling each annotator with a unique embedding, we accommodate their idiosyncrasies. By modeling the annotations themselves, we capture annotators' tendencies and views for individual annotation items.

To test our methods, we propose **TID-8**, **T**he **I**nherent **D**isagreement - **8** dataset, a benchmark that consists of eight existing language understanding datasets that have inherent annotator disagreement. TID-8 covers the tasks of NLI, sentiment analysis, hate speech detection, and humorousness comparison. Empirical results on TID-8 show that annotator embeddings improve performance on tasks where individual differences such as the sense of humor matter, while annotation embeddings give rise to clusters, suggesting that annotation embeddings aggregate annotators with similar annotation behaviors.

Our approach helps models learn significantly better from disagreements on six datasets in TID-8 and yields a performance gain between 4%~17% on four datasets that contain more than 50 annotators, while adding fewer than 1% of model parameters. We also conduct an ablation study and a comparison of the two embeddings over different datasets. By building and analyzing embeddings specific to the viewpoints of different annotators, we highlight the importance of considering annotator and annotation preferences when constructing models on data with disagreement. We hope to contribute towards democratizing AI by allowing for the representation of a diverse range of perspectives and experiences.

In summary, our contributions include:

- Rather than aggregating labels, we propose a setting of training models to directly learn from data that contains inherent disagreements.

- We propose **TID-8**, **T**he **I**herent **D**isagreement - **8** dataset, a benchmark that consists of eight existing language understanding datasets that have inherent annotator disagreements.

- We propose weighted annotator and annotation embeddings, which are model-agnostic and improve model performances on six out of the eight datasets in TID-8.

- We conduct a detailed analysis on the performance variations of our methods and how our

methods can be potentially grounded to real-world demographic features.

TID-8 is publically available on Huggingface at `https://huggingface.co/datasets/dnaihao/TID-8`. Our code and implementation are available at `https://github.com/MichiganNLP/Annotator-Embeddings`.

## 2 Related Work

**Inherent Annotator Disagreement.** Annotator disagreement is a well-known issue in NLP. A common approach to deal with annotator disagreement is to aggregate labels by taking the average (Pavlick and Callison-Burch, 2016) or the majority vote (Sabou et al., 2014), or select a subset of the data with a high annotator agreement rate (Jiang and de Marneffe, 2019a,b).

Researchers have criticized the conventional approach of assuming a single ground truth and ignoring the inherent annotator disagreement (Plank, 2022). Various studies reveal that there exists genuine human variation in labeling because of the subjectivity of the task or multiple plausible answers (Passonneau et al., 2012; Nie et al., 2020; Min et al., 2020; Ferracane et al., 2021; Jiang and Marneffe, 2022). For instance, in the task of toxic language detection, not all text is equally toxic for everyone (Waseem, 2016; Al Kuwatly et al., 2020). The identities and beliefs of the annotator influence their view toward the toxic text (Sap et al., 2022). Therefore, such annotator disagreement should not be simply dismissed as annotation "noise" (Pavlick and Kwiatkowski, 2019). Recently, researchers have started to leverage the different labels from annotators to better personalize the model for various users (Plepi et al., 2022).

**Modeling Annotator Disagreement.** Researchers have proposed various approaches for studying datasets with annotator disagreement. Zhang and de Marneffe (2021) propose Artificial Annotators to simulate the uncertainty in the annotation process. Zhou et al. (2022) apply additional distribution estimation methods such as Monte Carlo (MC) Dropout, Deep Ensemble, Re-Calibration, and Distribution Distillation to capture human judgment distribution. Meissner et al. (2021) train models directly on the estimated label distribution of the annotators in the NLI task. Zhang et al. (2021) consider annotator disagreement in a more general setting with a mixture of single-label, multi-label, and unlabeled

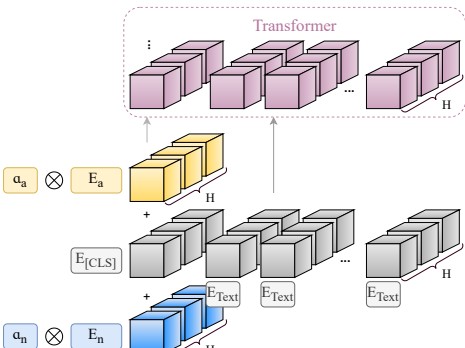

Figure 1: Concept diagram of how we apply our methods. We add the weighted annotator and annotation embeddings to the embedding of the "[CLS]" token, and feed the updated embedding sequence to the transformer-based models.

examples. Gordon et al. (2022) introduce jury learning to model every annotator in a dataset with Deep and Cross Network (DCN) (Wang et al., 2021). They combine the text and annotator ID together with the predicted annotator's reaction from DCN for classification. In contrast, we propose to explicitly embed annotator and their labels, and we perform a detailed analysis of these two embeddings. Davani et al. (2022) employ a common shared learned representation while having different layers for each annotator. Similar to our work, Kocoń et al. (2021) also develop trainable embeddings for annotators. In contrast, we propose embedding annotators as well as their labels with learnable matrices associated with each. We test our methods on eight datasets sourced from various domains, while Kocoń et al. (2021) conduct their experiments on four datasets all sourced from Wikipedia.

## 3 Task Setup

For the dataset $\mathcal{D}$, where $\mathcal{D} = (x_i, y_i, a_i)_{i=1}^{E}$, $x_i$ represents the input text, $y_i$ represents the corresponding label assigned by the annotator $a_i$ for text $x_i$. The dataset $\mathcal{D}$ consists of $E$ examples annotated by $N$ unique annotators. We aim to optimize the model parameters $\theta$ to maximize the likelihood of the correct labels given the annotator $a_i$ and all of their input text $x_i$:

$$\theta^* = \arg\max_\theta \sum_{i=1}^{E} \log P(y_i | x_i, a_i; \theta)$$

## 4 Methods

To explicitly account for annotation idiosyncrasies, we propose to create representations for both annotators and annotations. We create two embeddings, annotator embeddings ($E_a$), and annotation embeddings ($E_n$), associated with two learnable matrices ($\alpha_a$, $\alpha_n$), respectively as shown in Figure 1. For the annotator embeddings, we assign each annotator a unique embedding that represents their individual annotating preferences. For the annotation embeddings, we first assign embeddings to each label in the dataset. We then take the average embedding of the labels annotated by an annotator on other examples as their annotation embedding. The intuition is that an annotator's labels on other examples can be viewed as a proxy of their annotation tendencies when they annotate the current example. We describe the two embedding methods in detail below.

### 4.1 Embeddings

**Annotator Embedding ($E_a$)** We define a learnable matrix $E_A \in R^{N \times H}$ to represent embeddings for all the annotators, where $N$ is the total number of annotators, and $H$ is the hidden size of the model. The annotator embedding for an individual annotator is $E_a \in R^{1 \times H}$.

**Annotation Embedding ($E_n$)** We define a learnable matrix $E_L \in R^{M \times H}$ to represent embeddings for all the labels, where $M$ is the number of possible labels within the benchmark, and $H$ is the hidden size of the model. The embedding for an individual label $l$ is $E_l \in R^{1 \times H}$. During training, for the example $\kappa$ annotated by annotator $i$, we calculate the annotation embedding $E_n$ by taking the average of the label embeddings $E_l$ for all other examples annotated by the same annotator $i$:

$$E_n = \frac{1}{|K_i| - 1} \sum_{k \in K_i \setminus \{\kappa\}} E_{l(k)}$$

where $K_i$ is the set of examples in the training set annotated by the annotator $i$, the cardinality symbol $|\cdot|$ yields the number of elements within that set, and $E_{l(k)}$ indicates the embedding for label $l$ assigned to example $k$.

During testing, we average all the annotation embeddings of the training examples annotated by the same annotator:

$$E_n = \frac{1}{|K_{i,\text{train}}|} \sum_{k \in K_{i,\text{train}}} E_{l(k)}$$

## 4.2 Embedding Weights

The annotator and annotation embeddings are integrated into a transformer-based classification model. First, we calculate the sentence embedding of the input text $E_s \in R^{1 \times H}$ by averaging the text embedding, $E_t \in R^{\mathcal{T} \times H}$ over the number of tokens, $\mathcal{T}$ by Equation (1), where $E_t$ is the sum of the word embedding, type embedding, and position embedding from the original BERT embeddings.

$$E_s = \frac{1}{\mathcal{T}} \sum_{t=1}^{\mathcal{T}} (E_t)_{t,H} \qquad (1)$$

To incorporate our embedings, given the sentence embedding $E_s \in R^{1 \times H}$ and the annotator embedding $E_a \in R^{1 \times H}$, we calculate the weight for the annotator embedding $\alpha_a \in R^{1 \times 1}$ using Equation (2), where $W_s \in R^{H \times H}$ and $W_a \in R^{H \times H}$ are learnable matrices.

$$\alpha_a = (W_s E_s^T)^T (W_a E_a^T) \qquad (2)$$

Similarly, for the sentence embedding $E_s \in R^{1 \times H}$ and the annotation embedding $E_n \in R^{1 \times H}$, we calculate the weight for the annotation embedding $\alpha_n \in R^{1 \times 1}$ using Equation (3), where $W_n \in R^{H \times H}$ is another learnable matrix.

$$\alpha_n = (W_s E_s^T)^T (W_n E_n^T) \qquad (3)$$

We experiment with the following three methods for defining $\mathbf{E}$, the combined embedding used by the classification model:

$\mathbf{E_n}$: Text embedding and weighted annotator embedding. $\mathbf{E} = \{E_{[CLS]} + \alpha_n E_n, E_{t,1}, \cdots, E_{t,\mathcal{T}}\}$, where $E_{[CLS]}$ is the embedding of the first token, [CLS], the encoded representation of which is used for classification.

$\mathbf{E_a}$: Text embedding and weighted annotation embedding. $\mathbf{E} = \{E_{[CLS]} + \alpha_a E_a, E_{t,1}, \cdots, E_{t,\mathcal{T}}\}$.

$\mathbf{E_n + E_a}$: Text, weighted annotator, and weighted annotation embedding. $\mathbf{E} = \{E_{[CLS]} + \alpha_n E_n + \alpha_a E_a, E_{t,1}, \cdots, E_{t,\mathcal{T}}\}$.

The embedding $\mathbf{E}$ then propagates through the layer norm and the dropout function in the same way as the standard embedding calculation in the transformer-based model. The output embedding then propagates to the encoder.

## 5 TID-8 Overview

We propose TID-8: The Inherent Disagreement - 8 dataset. TID-8 consists of eight publicly available classification datasets with inherent annotator disagreement. In addition, information on the association between annotators and labels is available for all the datasets in TID-8. TID-8 covers the tasks of natural language inference (NLI), sentiment and emotion classification, hate speech detection, and humorousness comparison.

### 5.1 Desiderata Dataset

When selecting datasets for TID-8, a major concern is the quality of the annotations. Although there is a significant number of annotator disagreements arising from differences in interpretation, certain preferences, difficult cases, or multiple plausible answers, annotation errors could still be the reason for disagreements (Plank, 2022). Furthermore, there is no easy way to determine whether a label is assigned by mistake or because of subjective reasons.

Fortunately, each dataset has its own quality control mechanisms, such as including control examples (De Marneffe et al., 2019), various data analyses (Demszky et al., 2020), etc. For instance, during the collection process of the CommitmentBank dataset, De Marneffe et al. (2019) constructed control examples to assess annotators' attention, where the control examples clearly indicated certain labels. De Marneffe et al. (2019) filtered data from annotators who gave other responses for the control examples. Appendix B.4 contains details of the quality control for each dataset.

### 5.2 TID-8 Overview

TID-8 consists of eight datasets described in Table 2.

**Annotation Distribution.** Figure 2 shows the annotation distributions for the datasets in TID-8. In Sentiment (SNT) dataset, each annotator labels a similar number of examples. In Go Emotions (GOE), CommitmentBank (COM), Humor (HUM), and MultiDomain Agreement (MDA), a small group creates most of the dataset examples, though more than two-thirds of the annotators annotate more than 2,000 examples in Go Emotions (GOE). In Friends QIA (FIA), HS-Brexit (HSB), and Pejorative (PEJ) datasets, there are only a few annotators who each annotates the entire dataset, except for one annotator in the Pejorative (PEJ)

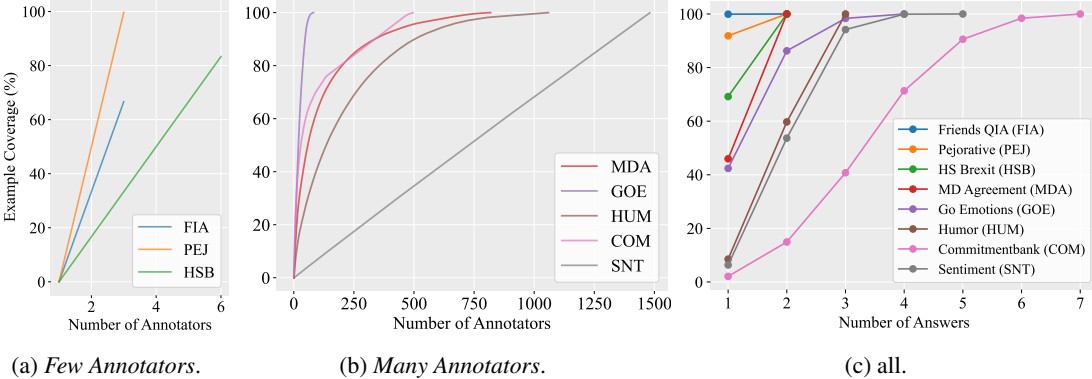

(a) *Few Annotators.*  (b) *Many Annotators.*  (c) all.

Figure 2: Figures 2a and 2b show the proportion of examples covered by the number of annotators (sorted by number of annotations). Specifically, Figure 2a shows the pattern for *Few Annotators Datasets* in TID-8 which contain < 10 annotators, while Figure 2b shows the pattern for *Many Annotators Datasets* in TID-8 which contain > 50 annotators. Figure 2c shows the proportion of examples with different numbers of labels on the eight datasets. The y-axis for all three plots is example coverage (%).

| | |
|---|---|
| **FIA** | Friends QIA (Damgaard et al., 2021) which classifies indirect answers to polar questions. |
| **PEJ** | Pejorative (Dinu et al., 2021) which classifies whether Tweets contain pejorative words. |
| **HSB** | HS-Brexit (Akhtar et al., 2021), an abusive language detection dataset. |
| **MDA** | MultiDomain Agreement (Leonardelli et al., 2021), a hate speech detection dataset. |
| **GOE** | Go Emotions (Demszky et al., 2020), an emotion classification dataset. |
| **HUM** | Humor (Simpson et al., 2019) which compares humorousness between a pair of texts. |
| **COM** | CommitmentBank (De Marneffe et al., 2019), an NLI corpus. |
| **SNT** | Sentiment Analysis (Díaz et al., 2018), a sentiment classification dataset. |

Table 2: Overview of datasets in TID-8. Appendix B.2 provides additional details for these datasets.

| Data | Train | Test | #A | #E/#A | #L |
|------|-------|------|-----|-------|-----|
| FIA | 4.4k | 0.6k | 3 | 1,873 | 5 |
| PEJ | 1.5k | 0.7k | 3 | 724 | 3 |
| HSB | 4.7k | 1k | 6 | 952 | 2 |
| MDA | 33k | 11k | 819 | 60 | 2 |
| GOE | 136k | 58k | 82 | 2,361 | 4 |
| HUM | 99k | 42k | 1059 | 133 | 3 |
| COM | 7.8k | 3.7k | 496 | 24 | 7 |
| SNT | 59k | 1.4k | 1481 | 41 | 5 |

Table 3: Statistics of datasets in TID-8. "#A" is the number of annotators, "#E/#A" is the average number of examples annotated per annotator, "#L" is the number of possible labels in the dataset. Datasets above the line are *Few Annotators Datasets* while datasets below are *Many Annotators Datasets*.

dataset who only annotates six examples. Therefore, we refer to FIA, HSB, and PEJ as *Few Annotators Datasets* as they contain fewer than 10 annotators. For comparison, we refer to the other datasets as *Many Annotators Datasets*, as all of them are annotated by more than 50 annotators. Appendix B.5 provides more details of the number of examples annotated for each annotator.

**Label Disagreement**  Figure 2c shows the label distributions among the datasets in TID-8. For most datasets, the majority of the examples have ≤ 3 possible labels. For CommitmentBank (COM), a significant proportion of the examples have four

or more labels. This aligns with the findings by Pavlick and Kwiatkowski (2019) that there are inherent disagreements in people's judgments in natural language inference tasks, especially considering the meticulous data collection process described in Section 5.1 that ensures high-quality and reliable datasets. Appendix B.6 provides more details of the number of examples corresponding to different numbers of answers.

## 6 Experiment Setup

In this paper, we investigate the setting of training models to directly learn from data that has inherent annotator disagreements. Therefore, instead of aggregating the labels, we consider each annotation as a separate example. In other words, different labels may exist for the same text annotated by different annotators.

**Models.** As our methods are model-agnostic, we test our methods with various language understanding models, including base and large versions of BERT (Devlin et al., 2019), RoBERTa (Liu et al., 2019), and DeBERTa Version 3 (He et al., 2021). Adding annotator or annotation embeddings only increases fewer than 1% of the original parameters for each model. Appendix C.3 provides the details of the calculation.

**Baseline Models.** We include four baseline models to compare against:

- **Random**: Randomly select a label.
- **Majority Vote (MV_ind)**: Always choose the most frequent label assigned by an annotator.
- **Majority Vote (MV_macro)**: Always choose the overall most frequent label across annotators.
- **T**: Only feed the example text to the model.
- **Naive Concat (NC)**: Naively concatenate the annotator ID with the example text and feed the concatenated string to the transformer-based models.

**Evaluation Metrics.** We report exact match accuracy (EM accuracy) and macro F1 scores on annotator-specific labels.

**Dataset Split.** Table 3 shows the statistics for each dataset in TID-8. We split the data annotated by each annotator into a train and test set (and a dev set if the original dataset contains one), where the train and test set have the same set of annotators ("annotation split"). For Friends QIA, HS-Brexit, MultiDomain Agreement, and Sentiment Analysis datasets, we follow the split from the original dataset. For the rest, we split the data into a 70% train set and a 30% test set. Appendix B.3 provides some pre-processing we conduct.

Appendix C.1 provides more details of the experimental set-ups.

# 7 Results and Discussion

**Both proposed embeddings help them better learn from disagreement.** Table 4 shows that annotation and annotator embeddings improve accuracy scores on *Many Annotators Datasets* in TID-8 up to 17% compared to the question-only baselines. In addition, naively concatenating the annotator ID with the question text harms the model performance, suggesting the need for sophisticated methods. Furthermore, we see our methods consistently

| | T | NC | $E_n$ | $E_a$ | $E_n + E_a$ |
|---|---|---|---|---|---|
| MDA | 75.06 | 64.81 | **76.70** | 75.72 | 75.76 |
| | 75.67 | 65.88 | 76.13 | 74.67 | 74.97 |
| | 75.65 | 68.36 | 76.02 | 75.14 | 75.28 |
| | 76.24 | 69.73 | **77.18** | 75.99 | 75.68 |
| | 76.45 | 70.43 | **77.78** | 76.93 | 77.26 |
| | 76.38 | 73.02 | 77.22 | 74.75 | 77.19 |
| GOE | 63.04 | 60.88 | 68.49 | **69.98** | 69.90 |
| | 62.90 | 62.12 | 68.39 | **69.92** | 66.32 |
| | 63.22 | 60.49 | 67.42 | **69.22** | 68.54 |
| | 63.19 | 58.86 | 64.41 | 65.71 | **68.46** |
| | 63.59 | 62.28 | 68.58 | **69.70** | 69.60 |
| | 62.94 | 58.60 | 65.18 | 66.52 | **69.74** |
| HUM | 54.26 | 52.05 | 56.72 | **58.15** | 53.89 |
| | 54.11 | 51.07 | 56.67 | **58.19** | 54.35 |
| | 54.43 | 47.16 | 55.07 | **56.31** | 53.31 |
| | 54.40 | 52.55 | 54.26 | 51.97 | 50.02 |
| | 54.71 | 53.63 | 56.33 | **57.70** | 53.31 |
| | 54.67 | 54.81 | 57.18 | **58.76** | 51.86 |
| COM | 40.83 | 40.78 | 44.00 | 44.22 | **44.41** |
| | 40.47 | 40.08 | 43.11 | **44.09** | 43.86 |
| | 41.44 | 41.61 | 40.81 | 42.62 | **43.00** |
| | 40.66 | 40.32 | 40.42 | 40.34 | 39.75 |
| | 40.54 | 38.14 | 42.37 | **42.82** | 42.59 |
| | 40.57 | 33.82 | 44.02 | **44.33** | 44.15 |
| SNT | 47.09 | 39.20 | 62.88 | 60.23 | **64.61** |
| | 47.32 | 36.91 | 61.88 | 56.20 | **63.65** |
| | 46.40 | 43.32 | **60.30** | 45.57 | 59.65 |
| | 47.88 | 43.82 | **58.19** | 46.50 | 55.16 |
| | 45.75 | 43.62 | **61.21** | 52.57 | 60.83 |
| | 48.76 | 43.78 | 67.37 | 68.39 | **69.77** |

Table 4: EM accuracy scores for annotation split on *Many Annotator Datasets*, averaged across 10 runs. The best results (statistically significant from baselines, t-test, p ≤ 0.05) are shown in bold. For each dataset, the six rows (from top to bottom) correspond to the scores from BERT base, BERT large, RoBERTa base, RoBERTa large, DeBERTa V3 base, and DeBERTa V3 large. Table 9 in Appendix C.4 shows the complete table including the remaining three baseline models and performances on *Few Annotators Datasets*.

improve performance across different models and model sizes, which shows the effectiveness of annotator and annotation embeddings in helping models learn from crowd data that has disagreements. Appendix C.4 provides macro F1 scores.

**Despite increased performance, annotator and annotation embeddings have different effects.** In Table 4, we see that on MultiDomain Agreement (MDA), Go Emotions (GOE), and Humor (HUM), adding either annotation or annotator embedding yields the best performance across models. In contrast, on CommitmentBank (COM) and Sentiment Analysis (SNT), adding both embeddings yields the best or second-to-best performance. Although on these two datasets, annotator or annota-

tion embedding might perform the best for some models, adding both embeddings yields a similar score. Furthermore, on Humor (HUM), adding both embeddings yields a worse performance than the baseline, while on Sentiment Analysis (SNT), adding both embeddings for BERT models yields the best performance. This suggests that annotator and annotation embedding have different effects. Intuitively, annotator embeddings capture individual differences. From Figure 5, we observe clusters emerge from annotation embedding. Therefore, we hypothesize that annotation embeddings align with group tendencies. We further discuss the effects of these embeddings in Section 8.

**Scaling models up does not help models learn from disagreement.** On the *Many Annotators Datasets* in TID-8, we observe little to no performance difference between the base and large versions of each model. In some cases, the large models even underperform the base model, as seen with BERT on the Go Emotions (GOE) dataset. This result suggests that the increased capacity of the larger models does not necessarily translate into improved performance when dealing with datasets that exhibit significant annotation disagreement. However, when there is minimal disagreement, as observed in the Friends QIA (FIA) dataset where only four examples have multiple labels while the remaining 5.6k examples have a single label (as shown in Table 8), the larger models consistently outperform the base versions for the text-only baseline (as shown in Table 9 in Appendix C.4). This trend could be attributed to the larger models' higher capacity and increased number of parameters.

The superior performance of the larger models in low-disagreement scenarios suggests that they excel at capturing and leveraging subtle patterns present in the data. However, when faced with datasets that contain significant disagreement, the larger models may become more prone to overfitting. Their increased capacity and specificity to the training data might hinder their ability to generalize well to new or unseen examples, resulting in diminished performance.

**Models' ability to learn from disagreements is similar for text-only baselines but varies with different embeddings.** For *Many Annotators Datasets* in TID-8, we observe similar performances across models for text-only baselines. This suggests that these models possess a comparable

| *Text:* We know it anecdotally from readers we've heard from who've been blatantly discriminated against because they're older. | | | | |
|---|---|---|---|---|
| POSITIVE (2) <–> NEGATIVE (-2) | | | | |
| **Annotator ID** | 1 | 2 | 3 | 4 |
| **Gold** | -1 | 0 | -2 | -2 |
| **T** | -1 | -1 | -1 | -1 |
| $\mathbf{E}_n + \mathbf{E}_a$ | -1 | 0 | -1 | -2 |

Table 5: An example predicted by the BERT base model from Sentiment Analysis, where adding both annotator and annotation embeddings better accommodates annotators' preference.

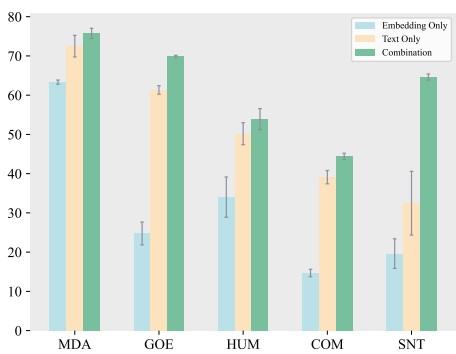

Figure 3: Ablation of performance on annotation split in the test when using annotator and annotation embeddings without text (Embedding Only), text embeddings (Text Only), or the combination (Combination). We use the BERT base model on *Many Annotators Datasets* in TID-8.

ability to learn from data that exhibits disagreements. However, the performance of these models varies when we incorporate annotation or annotator embeddings. This indicates that different pretraining strategies might have an impact on the effectiveness of incorporating annotator or annotation embeddings into a given model.

Apart from these analyses, we discuss performance patterns on each dataset in Appendix C.5.

## 8 Further Analyses

Since we observe a similar pattern across different transformer-based models, we use the BERT base model for ablation and discussion in this section.

**Our methods give annotator-based predictions.** Often, the baseline text-only model cannot accommodate different annotators, as shown in Table 5. However, after we incorporate the annotator or annotation embedding, the model can adjust its prediction to better align with the annotation for different annotators.

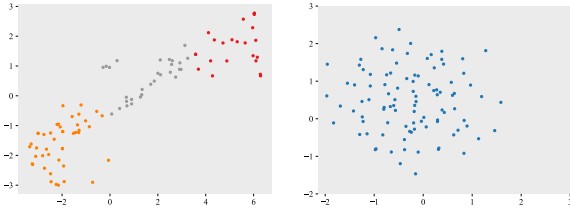

(a) Annotation Embedding     (b) Annotator Embedding

Figure 4: TSNE plots for MultiDomain Agreement. The embeddings are learned with BERT base model. We try with various hyperparameters and all of the plots demonstrate similar patterns.

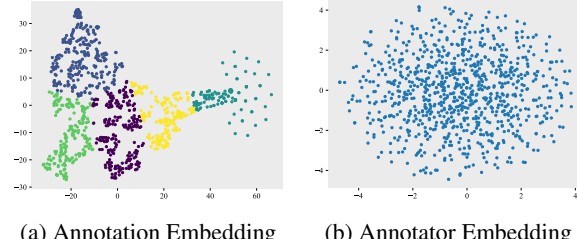

(a) Annotation Embedding     (b) Annotator Embedding

Figure 5: Annotation and annotator embedding for Sentiment Analysis. The embeddings are learned with the BERT base model. Different colors in Figure 5a indicate different "groups" in Table 17 in Appendix D.

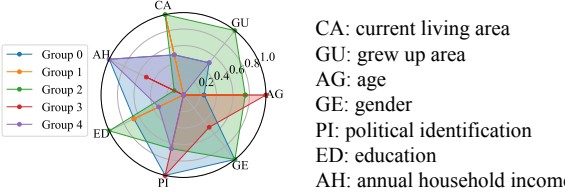

CA: current living area
GU: grew up area
AG: age
GE: gender
PI: political identification
ED: education
AH: annual household income

Figure 6: The prevalent demographic features for each cluster/group in Figure 5a in Sentiment Analysis (SNT). Appendix D provides details of these demographic features. Figure 10 provides the spread-out plots for each group.

**Text and annotator or annotation embeddings used jointly yield the best performance.** To reveal the performance contribution of different components, we train a BERT base model with text and both embeddings ($E_n + E_a$) and test with text, embeddings, or their combination separately. Figure 3 shows the test-time performance of using both embeddings (Embedding Only), just the text embeddings (Text Only), and using a combination of both (Combination). We can see that the $E_a + E_n$ and text embeddings need to work cooperatively to yield the best performance. In addition, we investigate the effects of the weight associated with the annotation or annotation embeddings in Appendix C.6.

**Annotator embeddings capture individual differences.** On Go Emotions (GOE) and Humor (HUM), adding annotator embeddings yields the best performance (Table 4). We hypothesize this is because both emotion and humor are subjective feelings, and annotator embeddings capture individual differences. As revealed by psychological studies, emotion and humor are entangled with one's cognition, motivation, adaptation, and physiological activity (Lazarus, 1991; Rowe, 1997; Tellegen et al., 1988; Cherkas et al., 2000; Martin and Ford, 2018). Having a dedicated embedding for each annotator (annotator embedding) might better capture the individual annotator differences in tasks dealing with emotion and humor.

**Annotation embeddings align with group tendencies.** The visualization of the embeddings on MultiDomain Agreement in Figure 4a reveals a spectrum in terms of annotation embeddings, where each point within the spectrum represents an individual. The spectrum encompasses individuals positioned at opposite ends, as well as others dispersed throughout. This could be explained by

the domain of the dataset. MultiDomain Agreement contains English tweets about Black Lives Matter, Elections, and Covid-19 (Appendix B.2). Regarding these topics, each annotator has their own political beliefs and attitudes towards these topics, and their annotation is a good reflection of their beliefs and attitudes. Therefore, the annotation embeddings may reflect the collective tendencies of individuals sharing similar political beliefs and attitudes.

Figure 5 visualizes both annotator and annotation embeddings on Sentiment Analysis. We notice that clusters emerge in annotation embeddings, indicating there are collective labeling preferences among the annotators on this dataset.

**Prevalent demographic features vary among clusters in annotation embeddings.** To gain insights into the relationship between annotation embeddings and demographic features, we perform a clustering analysis on the annotation embeddings of Sentiment Analysis as shown in Figure 5a. Specifically, we employ the K-means algorithm (Lloyd, 1982) and empirically choose K = 5 clusters based on visual observations from Figure 5a. Next, we map the identified clusters back to the corresponding demographic features provided by the dataset, as illustrated in Figure 6. The mapping

|       | T      | NC    | $E_n$ | $E_a$ | $E_n + E_a$ |
|-------|--------|-------|-------|-------|-------------|
| MDA   | **74.91** | 67.26 | 73.55 | 73.90 | 74.24 |
| GOE   | **62.86** | 61.01 | 61.33 | 61.98 | 61.96 |
| HUM   | **54.33** | 52.24 | 53.15 | 53.53 | 53.51 |
| COM   | 40.78  | 40.49 | 40.80 | 40.30 | 40.28 |
| SNT   | **43.93** | 38.56 | 36.99 | 40.82 | 37.90 |

Table 6: EM accuracy for the BERT base model on annotator split for *Many Annotators Datasets*, averaged across 10 runs. The best results are in bold if they yield a statistically significant difference from the baselines (t-test, $p \leq 0.05$).

is performed using the methods described in Appendix D. We find the prevalent demographic features associated with each cluster vary significantly. This analysis demonstrates that the learned annotation embeddings can be grounded in actual demographic features, allowing us to identify distinct patterns and tendencies within different clusters. By examining the relationships between annotation embeddings and demographic characteristics, we can gain a deeper understanding of the underlying dynamics at play in the dataset.

**Performance decrease is minimal on unknown annotators.** The annotation and annotator embeddings do not substantially degrade performance when testing on new annotators. We test the embeddings on the setting where the annotators in the train and test set are distinct ("annotator split"). We include 70% of the annotators in the train and 30% for the test in TID-8, and test with the BERT base model. Table 6 shows the EM accuracy scores for this annotator split. On most of the *Many Annotators Datasets* in TID-8, such as Go Emotions (GOE), MultiDomain Agreement (MDA), Humor (HUM), and CommitmentBank (COM), the performance loss is minimal to none. For Sentiment Analysis (SNT), the annotation embedding suffers a lot, which shows the difficulty of learning the group tendencies for unknown annotators. In addition, because sentiment and emotion are highly personalized feelings, annotator embeddings in this case suffer less than annotation embedding, as the annotator embeddings better handle individual differences. We further discuss performance on unknown annotators in Appendix C.4.

## 9 Conclusion

Instead of aggregating labels, we introduce a setting where we train models to directly learn from datasets that have inherent disagreements. We propose TID-8, The Inherent Disagreement - 8 dataset, consisting of eight language understanding tasks that have inherent annotator disagreement. We introduced a method for explicitly accounting for annotator idiosyncrasies through the incorporation of annotation and annotator embeddings. Our results on TID-8 show that integrating these embeddings helps the models learn significantly better from data with disagreements, and better accommodates individual differences. Furthermore, our approach provides insights into differences in annotator perspectives and has implications for promoting more inclusive and diverse perspectives in NLP models. We hope that TID-8 and our approach will inspire further research in this area and contribute to the development of more effective and inclusive NLP methods.

## Acknowledgement

We thank the anonymous reviewers for their feedback. We thank Zhenjie Sun, Yinghui He, and Yufan Wu for their help on the data processing part of this project. We also thank members of the Language and Information Technologies (LIT) Lab at the University of Michigan for their constructive feedback. This project was partially funded by an award from the Templeton Foundation (#62256).

## Limitations

One limitation of our work is the limited exploration of the demographic effects on annotations. This is because Sentiment Analysis is the only *Many Annotator Dataset* in TID-8 that provides publicly available demographic features for the annotators. We encourage researchers to collect the demographic features of the annotators when building datasets in the future while ensuring robust privacy protection.

Our methods suffer performance loss on unseen annotators, although on four of the five *Many Annotators Datasets* in TID-8, the performance loss is minimal to none. We stress that this is not the main focus of this paper, and we conduct this study to provide a more comprehensive understanding of the embeddings. Future studies might enhance methods to deal with annotator disagreement for "unseen" annotators.

Due to the scope of this project, we only studied annotator disagreement on classification tasks. However, annotator disagreement also exists in other NLP tasks such as summarization, or tasks beyond NLP such as image classification. We leave these topics to future research.

## Ethics Statement

In this paper, we do not collect any new datasets. Rather, we propose TID-8 based on eight publicly available datasets. All of the eight datasets in TID-8 provide the information of which annotator annotates the corresponding examples. In addition, HS-Brexit provides whether the annotator is a Muslim immigrant or not. Sentiment Analysis provides more comprehensive demographic features for each annotator. All annotator is anonymous and their personal information is not revealed.

Our methods require the information on which annotator annotates the corresponding examples, which we believe will accommodate annotators' preferences while protecting their personal information.

Furthermore, though we can observe that different prevalent demographic features vary for clusters in annotation embeddings, our methods do not rely on demographic features. This protects the privacy of the annotators.

Our methods help various models accommodate predictions based on annotators' preferences. We make steps towards leveraging different perspectives from annotators to enhance models.

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

## A  Author Contributions

Naihao Deng led the project, wrote the codebase, conducted all the experiments, and drafted the entire manuscript. Rada Mihalcea, Lu Wang, Siyang Liu, Xinliang Frederick Zhang provided suggestions, and helped with the brainstorming and paper writing refining. Siyang Liu provided her experience with experiment design, including improving baseline setting and significance testing, helped refine the writing in Section 5 and 7, and contributed significantly during the rebuttal period. Rada Mihalcea came up with the core idea of modeling annotators through their annotations, which was later refined in team discussions. Winston Wu provided feedback and suggestions for the project, refined writing, and contributed significantly during the rebuttal period.

## B  More about Selected Datasets in TID-8

### B.1  Dataset Examples

Table 7 shows examples in TID-8 where annotators disagree with each other.

### B.2  Selected Datasets

We provide the description of the eight datasets we selected for TID-8:

**FIA**  Friends QIA (Damgaard et al., 2021) is a corpus of classifying indirect answers to polar questions.

**PEJ**  Pejorative (Dinu et al., 2021) classifies whether Tweets contain words that are used pejoratively. By definition, pejorative words are words or phrases that have negative connotations or that are intended to disparage or belittle.

**HSB**  HS-Brexit (Akhtar et al., 2021) is an abusive language detection corpus on Brexit belonging to two distinct groups: a target group of three Muslim immigrants in the UK, and a control group of three other individuals.

**MDA**  MultiDomain Agreement (Leonardelli et al., 2021) is a hate speech classification dataset of English tweets from three domains of Black Lives Matter, Election, and Covid-19, with a particular focus on tweets that potentially lead to disagreement.

**GOE**  Go Emotions (Demszky et al., 2020) is a fine-grained emotion classification corpus of carefully curated comments extracted from Reddit. We

| **Friends QIA** | **Humor** |
|---|---|

**Friends QIA**

*Question:* Did Rachel tell you we hired a male nanny?

*Answer:* I think that's great!

ANN ANSWER (1), NOT THE ANSWER (2), ANSWER SUBJECT TO SOME CONDITIONS (3), NEITHER (4), OTHER (5): 1, 1, 4

**Humor**

*Text A:* Being crushed by large objects can be very depressing.

*Text B:* As you make your bed, so you will sleep on it.

ANN WHICH IS FUNNIER, X MEANS A TIE: A, A, B, X, X

**Pejorative**

*Text:* @WORSTRAPLYRICS Everything Jay-Z writes is trash.

ANN PEJORATIVE (1) <–> NON-PEJORATIVE (0): 1, 0, 0

**CommitmentBank**

*Premise:* Meg realized she'd been a complete fool. She could have said it differently. If she'd said Carolyn had borrowed a book from Clare and wanted to return it they'd have given her the address.

*Hypothesis:* Carolyn had borrowed a book from Clare.

ANN ENTAIL (3) <–>CONTRADICT (-3): 3, 3, 3, 2, 0, -3, -3, -3

**HS-Brexit**

*Text:* RT <user>: Islam has no place in Europe #Brexit.

ANN NO HATE (1) <–> HATE (0): 1, 1, 1, 0, 0, 0

**Sentiment Analysis**

*Text:* Even hotel bar food is good in California...fresh avocados, old chicken, and reasonably recent greens. Mmmm. Really.

ANN POSITIVE (2) <–>NEGATIVE (-2) : 2, 2, 0, -1

**MultiDomain Agreement**

*Text:* Please lost you yelling insanely at the sky on Nov 3 losers

ANN OFFENSIVE (1) <–> NOT OFFENSIVE (0): 1, 1, 1, 0, 0

**Go Emotions**

*Text:* This is how I feel when I use a crosswalk on a busy street

ANN POSITIVE (1), NEUTRAL (0), AMBIGUOUS (-1), NEGATIVE (-2): 1, 0

Table 7: Examples in TID-8 where annotators disagree with each other.

group emotions into four categories following sentiment level divides in the original paper.

**HUM** Humor (Simpson et al., 2019) is a corpus of online texts for pairwise humor comparison.

**COM** CommitmentBank (De Marneffe et al., 2019) is an NLI dataset. It contains naturally occurring discourses whose final sentence contains a clause-embedding predicate under an entailment canceling operator (question, modal, negation, antecedent of conditional). We note that, unlike the standard NLI setting where "Entailment", "Neutral", and "Contradiction" are used, we use the ratings from -3 to 3 that reflect the different degrees of "Entailment" and "Contradiction" and 0 to represent "Neutral" in particular.

**SNT** Sentiment Analysis (Díaz et al., 2018) is a sentiment classification dataset originally used to detect age-related sentiments.

### B.3 More Details about Dataset Pre-Processing

For annotation split on MultiDomain Agreement (MDA), we filtered 4746 examples (around 12% of the total examples) as the annotators of those do not appear in the training time for the annotation split. We use the entire dataset in the annotator split.

For Go Emotions (GOE), we follow Demszky

et al. (2020) to group the sentiments into positive, negative, ambiguous, and neutral.

### B.4 Quality Control from the Selected Datasets

**FIA** For Friends QIA, Damgaard et al. (2021) provide the raw agreement distribution of the annotations, which include the agreement for each of the final three categories. In addition, there is a high inter-annotator agreement rate as shown in Table 8.

**PEJ** For Pejorative, Dinu et al. (2021) recruited specialists in linguistics for the annotation. The Cohen's k agreement score was 0.933.

**HSB** For HS-Brexit, Akhtar et al. (2021) briefed and trained the annotators so that the annotators have a similar understanding of the abusive categories. For the purpose of their research, Akhtar et al. (2021) selected three volunteers who were first or second-generation immigrants and students from developing countries to Europe and the UK, of Muslim background; and three other volunteers who were researchers with western background and experience in the linguistic annotation.

**MDA** For MultiDomain Agreement, Leonardelli et al. (2021) selected a pool of tweets from the three domains of interest and asked three expert linguists to annotate them to ensure high-quality

annotations. The tweets with the perfect agreement are used as the gold standard. Leonardelli et al. (2021) then included a gold standard tweet in every HIT (group of 5 tweets to be annotated). If a crowd worker fails to evaluate the gold tweet, the HIT is discarded. Moreover, after the task completion, Leonardelli et al. (2021) removed all the annotations done by workers who did not reach a minimum overall accuracy of 70% with respect to the gold standard.

**GOE** For Go Emotions, Demszky et al. (2020) used a checkbox for raters to indicate if labeling a particular example was difficult, in which case raters could select "no emotions". Examples where no emotion was selected were removed from the dataset. In addition, Demszky et al. (2020) conducted various data analyses such as interrater correlation, correlation among emotions, principal preserved component analysis, etc. These various data analyses demonstrate the quality of the data, as the results are coherent and reasonable.

**HUM** For Humor, Simpson et al. (2019) computed the mean inter-annotator agreement (Krippendorff's $\alpha$) across instances. The result, 0.80, indicates a decent level of agreement among the annotators.

**COM** For CommitmentBank, De Marneffe et al. (2019) collected annotations through questionnaires that included eight discourses of interest and two constructed control discourses. These control discourses were used to evaluate the annotators' attentiveness, with one clearly indicating the speaker's certainty of the truth of the content of the complement (CC) and the other indicating certainty of the negation of the CC. Responses of +2 or +3 were accepted for the "true" control items, and responses of -3 or -2 were accepted for the "false" ones. Any data from annotators who gave other responses to at least one control item was excluded from the analysis.

**SNT** For Sentiment Analysis, Díaz et al. (2018) adopted the quality controls provided by Qualtrics[1], an annotation platform. Díaz et al. (2018) checked on completion time and whether there is straightlining. Specifically, the completion time check involved discarding any responses that were completed more quickly than half of the median speed. Note that the original paper did not talk about qual-

[1]https://www.qualtrics.com/iq/text-iq/

| Data | 1 | 2 | 3 | 4 | 5 | 6 + 7 |
|------|-----|------|------|-----|-----|-------|
| COM | 25 | 154 | 310 | 367 | 231 | 113 |
| SNT | 892 | 6.7k | 5.7k | 809 | 10 | |
| GOE | 24k | 25k | 7.0k | 941 | | |
| HUM | 2.4k | 14k | 11k | | | |
| MDA | 4.9k | 5.8k | | | | |
| HSB | 774 | 345 | | | | |
| PEJ | 856 | 76 | | | | |
| FIA | 5.6k | 4 | | | | |

Table 8: Label disagreements in TID-8.

ity control, we contacted the authors directly for the information.

### B.5 Annotation Distribution

Figure 7 shows the number of examples annotated by each annotator among the eight datasets. In Figure 7g, each annotator annotates a similar amount of examples. In Figures 7d to 7f and 7h, a small group creates most of the dataset examples similar to the pattern spotted by Geva et al. (2019), though more than 2/3 of the annotators annotate more than 2,000 examples in Figure 7h. In Figures 7a to 7c, there are just a few annotators and each annotates the entire datasets, except for one in Figure 7b who only annotates six examples.

### B.6 Label Distribution

Table 8 shows the number of examples corresponding to the number of unique labels of a single example in TID-8. Friends QIA is a dataset with little to no disagreement, as only 4 examples have two labels while the remaining 5.6k examples have a single label. Although there are examples in Sentiment Analysis and CommitmentBank where there is high disagreement, because of the rigorous quality control protocol described in Appendix B.4, we attribute them as hard examples or ambiguous examples that naturally lead to disagreement. We include all these examples in our modeling process.

## C More about Experiments

### C.1 Experiment Set-Ups

We adopt a learning rate of 1e-5 for all of our experiments. We set 3 epochs for TID-8. In terms of batch size, we find that a larger batch size helps stabilize the model performance. Therefore, we set the batch size from 8, 16, 32, 64, 128, and 256 based on the capacity of the GPUs. All of our experiments are run on the A40 GPU.

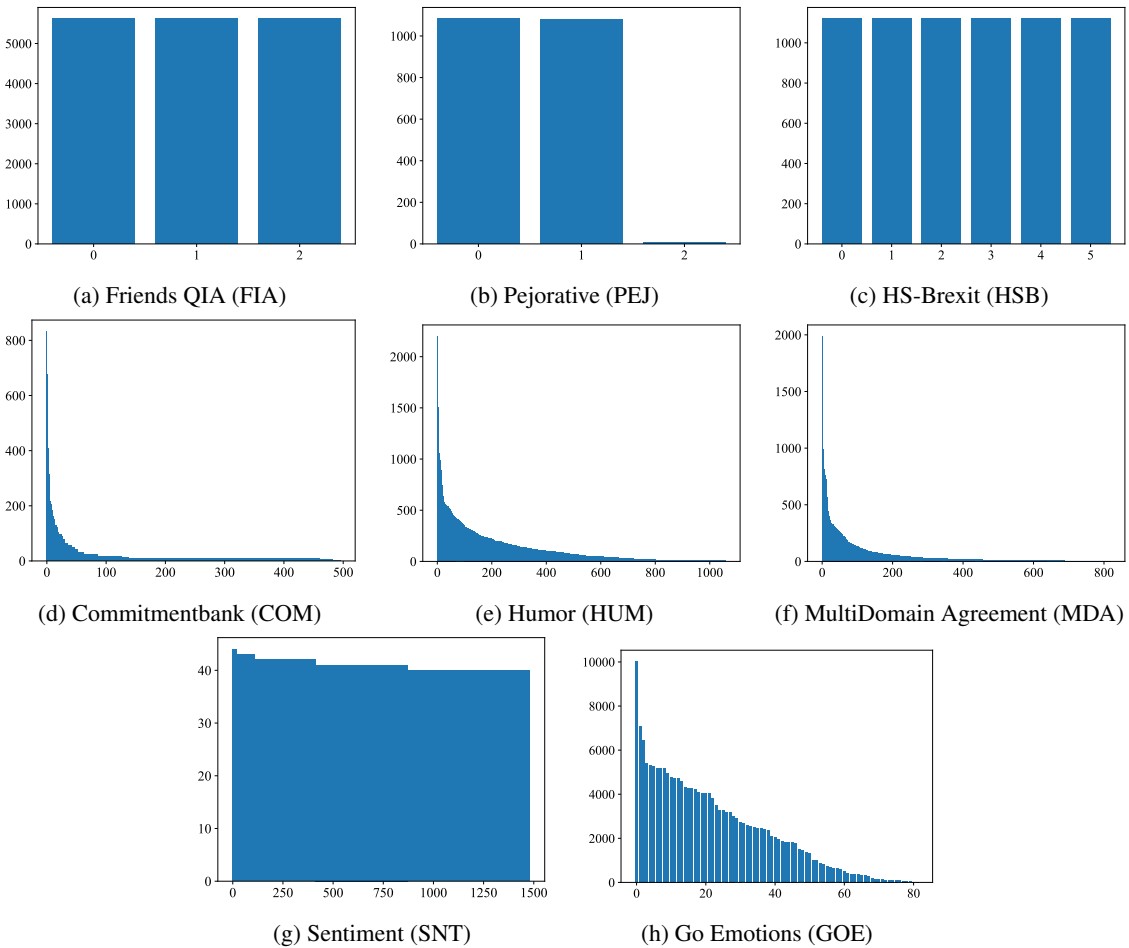

Figure 7: The number of examples annotated by each annotator in TID-8. We categorize the top three as *Few Annotators Datasets*, and the bottom five as *Many Annotators Datasets*.

## C.2 Baseline Models

Dawid and Skene (1979) introduced a probabilistic approach that allows to reduce the influence of unreliable annotators during predictions. Yet, the quality assurance methods in the TID-8 datasets ensure the labels are genuine and not spam. Instead of diminishing or consolidating labels, our interest lies in understanding how models can be trained on data with inherent disagreements. Consequently, we choose not to incorporate the models presented by Dawid and Skene (1979).

## C.3 Size of the Added Parameters

For the annotator embedding, the learnable matrix $E_A \in R^{N \times H}$ is of size $N \times H$, where $N$ is the number of annotators, $H$ is the hidden size of the model. For its associated weight, $\alpha_a$, we introduced $2 \times H \times H$ weight for the $W_s$ and $W_a$ matrices. Therefore, we introduced $NH + 2H^2$ parameters for the weighted annotator embedding. Similarly, we introduced $MH + 2H^2$ parameters

for the weighted annotation embedding, where $M$ is the number of unique labels in the dataset.

In TID-8, Sentiment Analysis has the most annotators of 1481, and CommitmentBank has the most unique labels of 7 (Table 3). Take the BERT base model as an example, $H = 768$, therefore, $NH + 2H^2 + MH + 2H^2 \approx 3H^2 + MH \approx$ 1Million at its maximum, which is around 1% of BERT base's parameter size (110 Million parameters).

## C.4 Experiment Results

Tables 9 and 10 report the EM accuracies and macro F1 scores across models for the annotation split on the eight datasets, respectively. We observe similar performance patterns between Tables 9 and 10.

We report the BERT base performance for the annotator split on the eight datasets in Tables 6, 11 and 12. Note that because for annotator split, we are testing on a different set of annotators from the train set, $MV_{ind}$ cannot make any prediction based

| | R | $MV_{ind}$ | $MV_{macro}$ | T | NC | $E_n$ | $E_a$ | $E_n + E_a$ |
|---|---|---|---|---|---|---|---|---|
| MDA | 50.03 | 61.71 | 63.58 | 75.06 | 64.81 | **76.70** | 75.72 | 75.76 |
| | | | | 75.67 | 65.88 | 76.13 | 74.67 | 74.97 |
| | | | | 75.65 | 68.36 | 76.02 | 75.14 | 75.28 |
| | | | | 76.24 | 69.73 | **77.18** | 75.99 | 75.68 |
| | | | | 76.45 | 70.43 | **77.78** | 76.93 | 77.26 |
| | | | | 76.38 | 73.02 | 77.22 | 74.75 | 77.19 |
| GOE | 25.05 | 41.27 | 36.71 | 63.04 | 60.88 | 68.49 | **69.98** | 69.90 |
| | | | | 62.90 | 62.12 | 68.39 | **69.92** | 66.32 |
| | | | | 63.22 | 60.49 | 67.42 | **69.22** | 68.54 |
| | | | | 63.19 | 58.86 | 64.41 | 65.71 | **68.46** |
| | | | | 63.59 | 62.28 | 68.58 | **69.70** | 69.60 |
| | | | | 62.94 | 58.60 | 65.18 | 66.52 | **69.74** |
| HUM | 33.30 | 45.65 | 41.55 | 54.26 | 52.05 | 56.72 | **58.15** | 53.89 |
| | | | | 54.11 | 51.07 | 56.67 | **58.19** | 54.35 |
| | | | | 54.43 | 47.16 | 55.07 | **56.31** | 53.31 |
| | | | | 54.40 | 52.55 | 54.26 | 51.97 | 50.02 |
| | | | | 54.71 | 53.63 | 56.33 | **57.70** | 53.31 |
| | | | | 54.67 | 54.81 | 57.18 | **58.76** | 51.86 |
| COM | 14.02 | 25.58 | 18.26 | 40.83 | 40.78 | 44.00 | 44.22 | **44.41** |
| | | | | 40.47 | 40.08 | 43.11 | **44.09** | 43.86 |
| | | | | 41.44 | 41.61 | 40.81 | 42.62 | **43.00** |
| | | | | 40.66 | 40.32 | 40.42 | 40.34 | 39.75 |
| | | | | 40.54 | 38.14 | 42.37 | **42.82** | 42.59 |
| | | | | 40.57 | 33.82 | 44.02 | **44.33** | 44.15 |
| SNT | 20.04 | 49.47 | 37.49 | 47.09 | 39.20 | 62.88 | 60.23 | **64.61** |
| | | | | 47.32 | 36.91 | 61.88 | 56.20 | **63.65** |
| | | | | 46.40 | 43.32 | **60.30** | 45.57 | 59.65 |
| | | | | 47.88 | 43.82 | **58.19** | 46.50 | 55.16 |
| | | | | 45.75 | 43.62 | **61.21** | 52.57 | 60.83 |
| | | | | 48.76 | 43.78 | 67.37 | 68.39 | **69.77** |
| FIA | 20.22 | 45.67 | 45.67 | 61.76 | 60.49 | 61.68 | 61.51 | 61.96 |
| | | | | 65.31 | 62.64 | 65.66 | 63.56 | 62.86 |
| | | | | **66.78** | 66.04 | 63.22 | 62.01 | 61.70 |
| | | | | 68.03 | 62.91 | 65.31 | 70.38 | 62.50 |
| | | | | 67.61 | 67.51 | 68.73 | 68.92 | 68.77 |
| | | | | 71.73 | 72.32 | 72.69 | 69.56 | 72.20 |
| PEJ | 33.78 | 51.90 | 51.90 | 67.48 | 64.84 | 65.28 | 65.42 | 65.77 |
| | | | | 68.59 | 67.22 | 64.93 | 62.84 | 62.94 |
| | | | | **71.46** | 71.20 | 61.29 | 62.32 | 60.82 |
| | | | | **71.93** | 70.39 | 59.89 | 57.66 | 59.23 |
| | | | | 70.26 | 63.19 | 64.70 | 64.69 | 65.07 |
| | | | | 74.51 | 73.98 | 73.20 | 73.05 | 72.91 |
| HSB | 50.04 | 86.90 | 86.90 | 86.87 | 86.01 | 86.90 | **87.80** | 87.68 |
| | | | | 86.35 | 85.75 | 86.61 | **87.83** | 87.10 |
| | | | | 86.77 | 86.65 | 86.61 | 87.03 | 86.69 |
| | | | | 86.83 | 87.04 | 86.68 | 85.76 | 87.22 |
| | | | | 86.90 | 86.71 | 87.16 | 87.78 | 87.87 |
| | | | | 86.87 | 86.31 | 87.49 | **88.75** | 88.04 |

Table 9: EM accuracy scores for annotation split on all eight datasets, where the same annotators appear in both train and test sets. We average the results across 10 runs. The best results are in bold if they yield a statistically significant difference from the baselines or the other way around (t-test, $p \leq 0.05$). For each dataset, the six rows correspond to the scores from BERT base, BERT large, RoBERTa base, RoBERTa large, DeBERTa V3 base, and DeBERTa V3 large.

| | R | $MV_{ind}$ | $MV_{macro}$ | T | NC | $E_n$ | $E_a$ | $E_n + E_a$ |
|---|---|---|---|---|---|---|---|---|
| MDA | 49.08 | 59.70 | 38.87 | 72.13 | 64.65 | **75.43** | 74.70 | 74.62 |
| | | | | 73.07 | 65.73 | **75.09** | 73.97 | 74.22 |
| | | | | 73.26 | 67.92 | **75.01** | 73.60 | 73.98 |
| | | | | 73.67 | 69.50 | **75.73** | 73.34 | 73.16 |
| | | | | 74.06 | 69.85 | **76.32** | 75.55 | 75.82 |
| | | | | 74.22 | 71.69 | **76.27** | 71.69 | 76.23 |
| GOE | 24.02 | 28.98 | 13.43 | 59.44 | 56.98 | 66.00 | **67.55** | 67.46 |
| | | | | 59.23 | 58.54 | 65.84 | **67.39** | 61.71 |
| | | | | 59.28 | 55.00 | 64.41 | **66.50** | 65.70 |
| | | | | 59.52 | 52.33 | 59.27 | 60.86 | **65.69** |
| | | | | 59.85 | 58.14 | 65.78 | **67.07** | 66.87 |
| | | | | 59.15 | 52.34 | 60.35 | 61.83 | **67.15** |
| HUM | 32.52 | 43.82 | 19.57 | 46.32 | 41.79 | 52.97 | **55.06** | 48.58 |
| | | | | 46.15 | 42.98 | 53.55 | **55.66** | 49.74 |
| | | | | 46.35 | 40.76 | 51.29 | **52.37** | 47.33 |
| | | | | 46.66 | 43.71 | 48.45 | 42.23 | 37.58 |
| | | | | 46.17 | 43.64 | 53.22 | **54.30** | 47.86 |
| | | | | 47.04 | 47.17 | 54.15 | **56.12** | 45.27 |
| COM | 13.86 | 22.15 | 4.41 | 32.21 | 31.30 | 36.38 | **36.87** | 36.76 |
| | | | | 31.41 | 31.39 | 35.02 | **37.39** | 36.64 |
| | | | | 32.32 | 32.60 | 31.85 | 33.42 | 33.41 |
| | | | | 32.34 | 32.29 | 32.13 | 31.10 | 30.82 |
| | | | | 29.71 | 25.92 | 31.89 | **32.78** | 32.50 |
| | | | | 30.72 | 23.29 | 36.64 | **38.71** | 37.93 |
| SNT | 18.29 | 38.85 | 10.91 | 32.71 | 28.39 | 58.16 | 48.99 | **59.40** |
| | | | | 34.30 | 30.63 | 57.40 | 43.78 | **58.68** |
| | | | | 34.40 | 32.90 | **55.40** | 33.64 | 54.19 |
| | | | | 35.45 | 33.54 | **50.47** | 33.30 | 46.22 |
| | | | | 33.58 | 27.78 | **55.72** | 40.16 | 55.11 |
| | | | | 37.45 | 36.08 | 64.36 | 64.11 | **67.14** |
| FIA | 17.07 | 12.54 | 12.54 | 45.22 | 44.91 | 44.30 | 45.65 | 45.36 |
| | | | | 49.06 | 42.80 | 47.23 | 44.23 | 45.55 |
| | | | | **50.82** | 48.40 | 46.19 | 44.75 | 45.58 |
| | | | | 51.24 | 41.15 | 44.92 | 56.94 | 41.12 |
| | | | | 49.04 | 53.02 | 52.32 | 52.78 | 53.68 |
| | | | | 55.75 | 56.95 | 56.49 | 53.72 | 56.87 |
| PEJ | 28.54 | 22.78 | 22.78 | 45.23 | 43.70 | 43.97 | 43.53 | 43.97 |
| | | | | 46.11 | 45.34 | 43.29 | 41.10 | 41.92 |
| | | | | **48.26** | 48.06 | 40.42 | 40.68 | 40.24 |
| | | | | **48.35** | 47.34 | 39.09 | 36.56 | 38.71 |
| | | | | 47.28 | 40.49 | 43.32 | 42.94 | 43.22 |
| | | | | 50.48 | 50.08 | 49.47 | 49.39 | 49.33 |
| HSB | 42.08 | 46.50 | 46.50 | 56.99 | 61.23 | 60.03 | 66.34 | 64.40 |
| | | | | 58.80 | 64.99 | 65.88 | 66.14 | **69.24** |
| | | | | 64.60 | 61.54 | 56.03 | 60.36 | 57.37 |
| | | | | 63.22 | 67.37 | 58.12 | 61.24 | 58.94 |
| | | | | 56.01 | 54.82 | 66.21 | **72.27** | 71.36 |
| | | | | 60.42 | 69.86 | 71.35 | **74.31** | 73.72 |

Table 10: Macro F1 scores for annotation split on all eight datasets, where the same annotators appear in both train and test sets. We average the results across 10 runs. The best results are in bold if they yield a statistically significant difference from the baselines or the other way around (t-test, $p \leq 0.05$). For each dataset, the six rows correspond to the scores from BERT base, BERT large, RoBERTa base, RoBERTa large, DeBERTa V3 base, and DeBERTa V3 large.

|  | R | MV$_{\text{macro}}$ | T | NC | E$_n$ | E$_a$ | E$_n$ + E$_a$ |
|---|---|---|---|---|---|---|---|
| MDA | 49.88 | 64.15 | **74.91** | 67.26 | 73.55 | 73.90 | 74.24 |
| GOE | 24.96 | 36.46 | **62.86** | 61.01 | 61.33 | 61.98 | 61.96 |
| HUM | 33.34 | 42.37 | **54.33** | 52.24 | 53.15 | 53.53 | 53.51 |
| COM | 14.22 | 19.88 | 40.78 | 40.49 | 40.80 | 40.30 | 40.28 |
| SNT | 19.98 | 42.56 | **43.93** | 38.56 | 36.99 | 40.82 | 37.90 |
| FIA | 20.22 | 46.41 | **84.55** | 83.00 | 77.48 | 83.70 | 83.23 |
| PEJ | 33.78 | 51.57 | **70.43** | 66.46 | 52.45 | 63.71 | 59.47 |
| HSB | 49.88 | 95.09 | 87.77 | 89.30 | 80.78 | 89.34 | 88.99 |

Table 11: EM accuracy for the BERT base model on annotator split, where a different set of annotators appear in train and test sets. We average the results across 10 runs. The best results are in bold if they yield a statistically significant difference from the baselines or the other way around (t-test, $p \leq 0.05$).

|  | R | MV$_{\text{macro}}$ | T | NC | E$_n$ | E$_a$ | E$_n$ + E$_a$ |
|---|---|---|---|---|---|---|---|
| MDA | 48.86 | 39.08 | 72.35 | 66.76 | 72.04 | 71.59 | 71.95 |
| GOE | 23.98 | 13.36 | 59.33 | 56.91 | 56.68 | 58.13 | 58.05 |
| HUM | 32.50 | 19.84 | 46.59 | 42.54 | **47.64** | 47.09 | 47.39 |
| COM | 13.97 | 4.74 | 32.08 | 31.31 | 32.58 | 32.12 | 31.81 |
| SNT | 17.92 | 11.94 | **31.18** | 32.82 | 23.99 | 29.35 | 26.82 |
| FIA | 17.07 | 12.68 | **75.59** | 73.27 | 66.42 | 73.70 | 72.62 |
| PEJ | 28.54 | 22.68 | **47.33** | 44.60 | 24.80 | 41.75 | 36.99 |
| HSB | 37.13 | 48.74 | 65.23 | 65.10 | 49.45 | 62.57 | 63.14 |

Table 12: Macro F1 scores for the BERT base model on annotator split, where a different set of annotators appear in train and test sets. We average the results across 10 runs. The best results are in bold if they yield a statistically significant difference from the baselines or the other way around (t-test, $p \leq 0.05$).

on its mechanism. Therefore, we omit the MV$_{\text{ind}}$ baseline in Tables 6, 11 and 12.

### C.5 Performances Patterns

Table 9 shows the EM accuracy scores in different settings on TID-8 for the annotation split. Here we mainly describe the performance patterns for the BERT base model. The improvement across different settings varies on TID-8. On CommitmentBank and Sentiment Analysis, adding either the annotator or annotation embeddings improves the model performance, and adding the two embeddings together further improves the model performance. On Go Emotions, HS-Brexit, and Humor, both embeddings improve the model performance, but adding the two embeddings yields less improvement than simply using annotator embeddings. On Multi-Domain Agreement, both annotator and annotation embeddings improve the model performance, but adding annotation embeddings yields the most performance gain. Additionally, adding both embeddings together yields less performance gain than annotation embedding only. On Pejorative, there are no significant improvements after adding the annotator or annotation embeddings. On Friends QIA, however, adding either embedding hurts the performance, and the baseline setting

achieves the best performance.

Table 10 in Appendix C.4 shows the macro F1 scores with similar trends.

**Our embeddings yield insignificant performance gains or performance losses on datasets with too few annotators or with few disagreements.** For Friends QIA and Pejorative, adding annotator or annotation embeddings yields insignificant performance gains or performance losses. For Friends QIA, every annotator annotates all of the examples in the dataset as shown in Figure 7. Moreover, only 4 examples in Friends QIA have 2 different labels, while the remaining 5.6k examples have only a single label as shown in Table 8. For Pejorative, the proportion of examples with which annotators disagree is relatively small, as less than 10% of examples have 2 distinct labels (Table 8). Moreover, Pejorative only has 3 annotators and 2 of them annotate the whole dataset while the other annotates fewer than 10 examples. In these scenarios with such a high or almost perfect agreement rate and so few annotators, adding extra annotator information may be a burden to the model, as there is not much the model can do to accommodate different annotators.

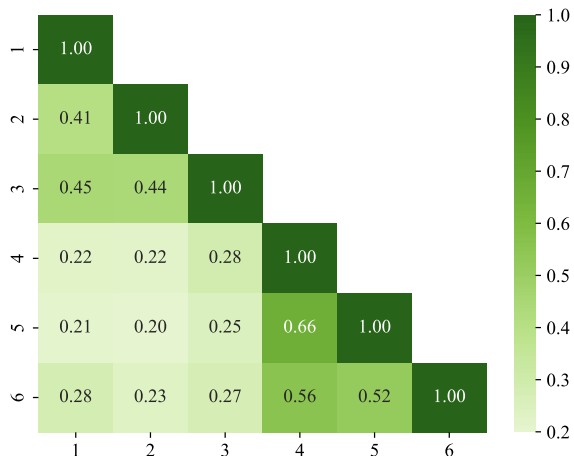

Figure 8: Cohen Kappa scores between each annotator of HS-Brexit. Annotators 4 to 6 are Muslim immigrants and 1 to 3 are not.

| NO HATE (NH) <–> HATE (H) | | | | | | |
|---|---|---|---|---|---|---|
| **Annotator ID** | 1 | 2 | 3 | 4 | 5 | 6 |
| **Group** | | Non Muslim | | Muslim Immigrants in the UK | | |
| *Text:* RT <user>: Islam has no place in Europe #Brexit <url> | | | | | | |
| **Annotation** | H | NH | H | NH | NH | H |
| *Text:* Who let this clown into the US? Deport now. <url> | | | | | | |
| **Annotation** | NH | H | NH | H | H | H |

Table 13: Label disagreements within the same demographic group for HS-Brexit. Annotator 4 to 6 are Muslim immigrants and 1 to 3 are not.

**Cases when annotator embeddings alone improve performances.** Apart from Go Emotions and Humor discussed in Section 8, on HS-Brexit, incorporating annotator embedding yields a performance gain. HS-Brexit is annotated by six annotators: three are Muslim immigrants in the UK, while the other three are not. As all of the annotators annotate the entire dataset, we are able to calculate inter-annotator agreement using the Cohen Kappa scores (McHugh, 2012) and examine the agreement between annotators belonging to the same group (Muslim immigrants or not). Figure 8 shows the Cohen Kappa scores, where annotators 4 to 6 are Muslim immigrants and 1 to 3 are not. Though the inter-group agreement is higher ($\geq 0.40$), both the inter-group and overall inter-annotator agreements lie in the range of 0.20 to 0.60, which suggests a fair or moderate agreement. Table 13 shows two examples where annotators from a Muslim background or no Muslim background disagree within their own groups. In such a case, annotator embedding might better capture the individual variance.

**Cases when annotation embeddings alone improve model performances.** As discussed in Section 8, on MultiDomain Agreement, adding annotation embeddings alone improves model performances. And the reason might be because the annotation is a good reflection of their political beliefs and attitudes towards topics in the dataset domains. Therefore, annotation embeddings may reflect the collective tendencies of individuals sharing similar political beliefs and attitudes.

Our findings align with social identity theory (Tajfel and Turner, 2004) which proposes that individuals within the same group exhibit similarities, while differences exist between groups due to variation in attitudes, behaviors, and self-concepts (Hewstone et al., 2002; Hogg, 2016; Rosenberg, 2017).

**Cases when adding annotator and annotation embeddings together yield the best performances.** For the Sentiment Analysis, adding both annotator and annotation embedding yields the best performance. The Sentiment Analysis dataset is an emotion classification dataset with the added specific goal of studying age-related bias, as shown in Tables 5 and 7. Apart from individual differences in emotional feelings, annotation embeddings can also capture tendencies as a group. Thus, considering both individual and group tendencies, we find that two embeddings together yield better results

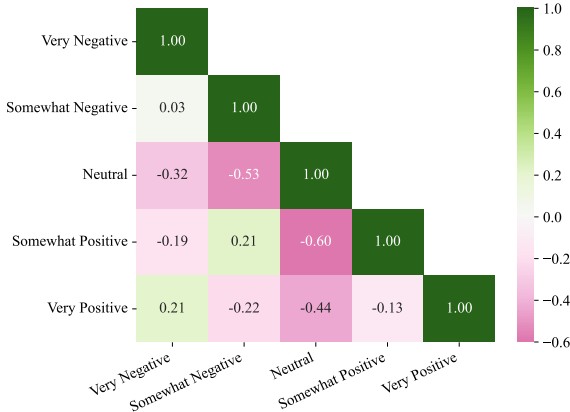

Figure 9: Pearson correlation of each label for Sentiment Analysis. There is a weak positive relationship between "very negative" and "very positive", as well as "somewhat negative" and "somewhat positive".

than using one alone on three out of the six models we tested.

Another reason why the group tendency is important for Sentiment Analysis is that it has fine-grained labels that indicate different intensities of the same feelings or judgments. For instance, unlike Go Emotions, which has three labels indicating positive, negative, or neutral, Sentiment Analysis has five labels to represent different extents of positive and negative emotion. Therefore, certain groups of annotators may have their own interpretations or preferences of the scale on Sentiment Analysis. We calculate the Pearson correlations across labels by:

- We first obtain the matrix of for annotators ($A_i$) who annotate more than 50 examples:

$$
\begin{array}{ccccc}
& \mathbf{A}_1 & \mathbf{A}_2 & \cdots & \mathbf{A}_N \\
\mathbf{label}_1 & v_{11} & v_{12} & \cdots & v_{1N} \\
\mathbf{label}_2 & v_{21} & v_{22} & \cdots & v_{2N} \\
\cdots & & & & \\
\mathbf{label}_5 & v_{51} & v_{52} & \cdots & v_{5N}
\end{array}
$$

- We then calculate the Pearson correlation scores based on the row vectors. For instance, if we want to calculate the Pearson correlation score for $label_1$ and $label_2$, we would calculate with respect to $[v_{11}, v_{12}, \cdots, v_{1N}]$ and $[v_{21}, v_{22}, \cdots, v_{2N}]$.

Because the examples are randomly assigned by default, we assume that there would not be any obvious correlation between labels. However, Figure 9 shows a moderate Pearson correlation score

for the "Somewhat Negative" and "Somewhat Positive" labels. This suggests that a group of annotators may prefer to use the a "moderate" extent in their labeling process.

## C.6 The Effects of Weights in Annotator and Annotation Embeddings

Tables 14 and 15 present a comparison between the weighted annotator and annotation embeddings versus the embeddings that are without the weight matrix.

We notice that for the eight datasets, the weighted version generally performs better or about the same as its unweighted counterpart. The weight $\alpha_{\mathbf{a}}, \alpha_{\mathbf{n}}$ may capture the relations between the text and the annotator or annotation embeddings, therefore the weighted annotator or annotation embeddings may be integrated better with the text embeddings.

One exception is the accuracy patterns for MultiDomain Agreement. However, the weighted embeddings still achieve better or similar macro F1 scores on MultiDomain Agreement.

The other exception is Friends QIA in *Few Annotators Datasets*, where the unweighted embeddings outperform the weighted ones. The reason could be that Friends QIA only has six annotators and the disagreement is minimal. Therefore, the unweighted embeddings alone may be enough to capture individual preferences.

## C.7 Take-Away Messages

**Single-dimensioned demographic features are not enough.** We find that individuals belonging to the same demographic groups may hold contrasting opinions. For instance, on HS-Brexit, where the text might contain hate speech towards the Muslim community, being from the Muslim community or not does not necessarily determine an individual's stance or opinion. Table 13 shows several examples where people from the same cultural background disagree with each other. Our findings are similar to Biester et al. (2022), who studied annotation across genders for datasets of sentiment analysis, natural language inference, and word similarity, and found a lack of statistically significant differences in annotation by males and females on three out of the four datasets. Thus, relying on a single demographic feature for analyzing perceptions oversimplifies the intricate nature of individual stances and opinions. We advocate for considering multiple dimensions to gain a more comprehensive understanding of

|     | $E_{\mathbf{n}}$w.o. weight | $E_{\mathbf{n}}$w. weight | $E_{\mathbf{a}}$w.o. weight | $E_{\mathbf{a}}$w. weight | $E_{\mathbf{n}}+E_{\mathbf{a}}$w.o. weight | $E_{\mathbf{n}}+E_{\mathbf{a}}$w. weight |
|-----|------|------|------|------|------|------|
| MDA | 75.17 | **76.70** | **76.86** | 75.72 | **77.19** | 75.76 |
| GOE | 66.68 | **68.49** | 69.00 | **69.98** | 69.20 | **69.90** |
| HUM | 54.13 | **56.72** | 55.37 | **58.15** | 54.65 | 53.89 |
| COM | 41.20 | **44.00** | 40.89 | **44.22** | 41.10 | **44.41** |
| SNT | 57.57 | **62.88** | 46.08 | **60.23** | 57.96 | **64.61** |
| FIA | 61.86 | 61.68 | 61.68 | 61.51 | 61.35 | 61.96 |
| PEJ | 66.43 | 65.28 | 65.78 | 65.42 | 66.67 | 65.77 |
| HSB | 87.03 | 86.90 | 87.19 | **87.80** | 87.25 | 87.68 |

Table 14: EM accuracy scores for our embeddings with or without weights on annotation split. We obtain the results from the BERT base model. We average the results across 10 runs. The best results are in bold if they yield a statistically significant difference from the baselines (t-test, p ≤ 0.05).

|     | $E_{\mathbf{n}}$w.o. weight | $E_{\mathbf{n}}$w. weight | $E_{\mathbf{a}}$w.o. weight | $E_{\mathbf{a}}$w. weight | $E_{\mathbf{n}}+E_{\mathbf{a}}$w.o. weight | $E_{\mathbf{n}}+E_{\mathbf{a}}$w. weight |
|-----|------|------|------|------|------|------|
| MDA | 74.88 | 74.70 | 72.42 | **75.43** | 75.34 | 74.62 |
| GOE | 66.47 | **67.55** | 63.57 | **66.00** | 66.71 | **67.46** |
| HUM | 48.96 | **55.06** | 45.69 | **52.97** | 47.54 | 48.58 |
| COM | 31.43 | **36.87** | 31.53 | **36.38** | 31.81 | **36.76** |
| SNT | 32.63 | **48.99** | 47.99 | **58.16** | 48.67 | **59.40** |
| FIA | 44.59 | 45.65 | 44.81 | 45.36 | 45.61 | 44.30 |
| PEJ | 44.71 | 43.97 | 44.08 | 43.53 | 44.80 | 43.97 |
| HSB | 55.48 | **66.34** | 54.30 | 60.03 | 54.80 | **64.40** |

Table 15: Macro F1 scores for our embeddings with or without weights on annotation split. We obtain the results from the BERT base model. We average the results across 10 runs. The best results are in bold if they yield a statistically significant difference from the baselines (t-test, p ≤ 0.05).

diverse viewpoints.

**Diversifying the data.** Because of the inherent individual differences, it is crucial to incorporate diversity in the data collection process. Collecting data from a wide range of sources, including individuals from diverse backgrounds and demographics, is imperative. There could be disagreements involved in the process, as we have seen in the eight datasets we studied in this paper. However, by gathering annotations from diverse populations, we can capture the richness and complexity of human experiences, perceptions, and opinions. Failing to account for these individual differences in data collection could lead to biased or incomplete representations, limiting the validity and generalizability of research findings.

### C.8  Performance of Our Methods on Other Datasets

Kumar et al. (2021) introduced the Toxic Ratings dataset. We obtained the complete dataset from Kumar et al. (2021) since the version available to the public lacks annotator identification. Table 16 shows the performance of our methods with the BERT base model.

## D  Details of the Group Alignment with Demographic Features

**Dimension of Demographic Features.** We select seven dimensions that are sequential or can be regarded as sequential data here, including age, grew-up area, current living area, annual household income, education, political identification, and gender. Table 17 shows examples of the different demographic features for each dimension. Note that the values for each dimension are the most prevalent ones in the group, they are not necessarily coming from a single person in that group.

**Methods** We first conduct the K-means clustering to cluster the embeddings. For an annotator $a_i$ corresponding to the data point in the cluster, we find his or her demographic features for all the dimensions in the metadata for each dimension $\{d_1, d_2, \cdots, d_{12}\}$. There could be imbalances for the number of people from $d_j$ corresponding to the dimension $D_j$, for instance, there might be more females than males in terms of the gender dimension. Therefore, when counting its frequency in a cluster/group, we give a multiplier $\alpha$ for $d_j$

$$\alpha = \frac{N}{\sum_{k=1}^{N} [\![ D_j = d_j ]\!]} \quad (4)$$

| | R | $MV_{ind}$ | $MV_{macro}$ | T | NC | $E_n$ | $E_a$ | $E_n + E_a$ |
|---|---|---|---|---|---|---|---|---|
| TOR | 20.06 | 55.39 | 52.36 | 53.17 | 48.08 | 58.07 | 57.37 | **58.63** |

Table 16: Performance of our methods on Toxic Ratings (TOR) dataset (Kumar et al., 2021) for annotation split.

, where $N$ is the total number of annotators (every annotator has their own demographic features), $[\![\,]\!]$ returns 1 if the dimension $D_j$ for that annotator is $d_j$, otherwise, it returns 0.

We then calculate the frequency of $d_j$ for the demographic dimension $D_j$ for each cluster, where the frequency $f$ for the demographic feature $D_j = d_j$ in cluster $c$ is

$$f = \alpha[\![D_j = d_j]\!]_c \qquad (5)$$

We show the most prevalent demographic feature for dimension $D_j$ for each cluster in Figures 6 and 10a to 10e, and list the demographic features for each cluster in Table 17.

**Normalization of Demographic Features**

- **Current Live Area**: Rural: 0.0, Suburban: 0.5, Urban: 1.0.

- **Grew Up Area**: Rural: 0.0, Suburban: 0.5, Urban: 1.0.

- **Age**: 50-59: 0, 60-69: 1, 70-79: 2, 80-89: 3, 90-99: 4, 100+: 5.

- **Gender**: Female: 0.0, Nonbinary: 0.5, Male: 1.0.

- **Political Identification**: Very liberal: 0.0, Somewhat liberal: 0.25, Moderate: 0.5, Somewhat conservative: 0.75, Very conservative: 1.0.

- **Education**: Less than high school: 0.0, High school graduate, GED, or equivalent: 0.25, Some college or associate's degree: 0.5, Bachelor's degree: 0.75, Graduate or professional degree: 1.0.

- **Annual Household Income**: Less than $10,000: 0.0, $10,000 - $14,999: 0.11, $15,000 - $24,999: 0.22, $25,000 - $34,999: 0.33, $35,000 - $49,999: 0.44, $50,000 - $74,999: 0.56, $75,000 - $99,999: 0.67, $100,000 - $149,999: 0.78, $150,000 - $199,999: 0.89, More than $200,000: 1.0.

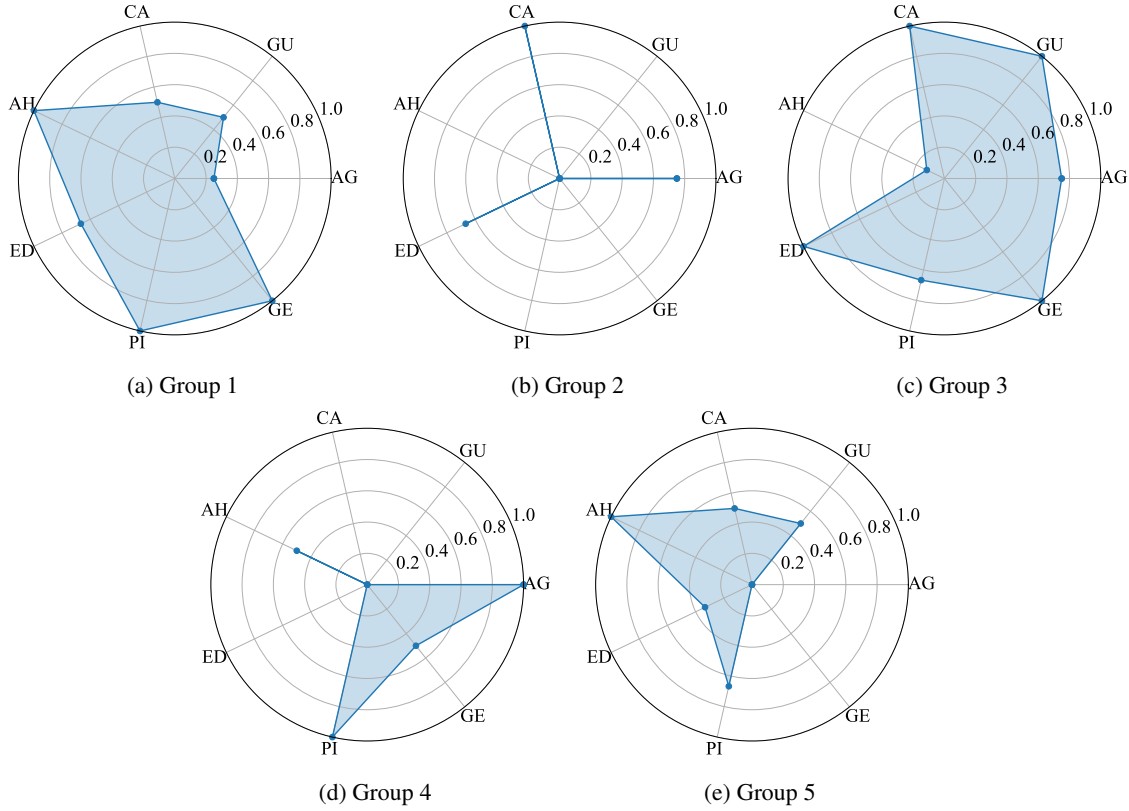

Figure 10: Sentiment Analysis group alignment with demographic features. For the shorthands, CA: current living area, GU: grew up area, AG: age, GE: gender, PI: political identification, ED: education, AH: annual household income.

| Group ID | 0 | 1 | 2 | 3 | 4 |
|---|---|---|---|---|---|
| **Curr Area** | Suburban | Urban | Urban | Rural | Suburban |
| **Grew area** | Suburban | Rural | Urban | Rural | Suburban |
| **Age** | 70-79 | 90-99 | 90-99 | 100+ | 60-69 |
| **Gender** | Male | Female | Male | Nonbinary | Female |
| **Poli Identifi** | Somewhat Conservative | Very liberal | Moderate | Somewhat Conservative | Moderate |
| **Edu** | College/Associate | College/Associate | Bachelor's degree | Less than high school | High School / GED / equiv--alent |
| **Annual Income** | $150k - $200k | < $10k | $10k - $15k | $35k - $50k | $150k - $200k |

Table 17: The most prevalent demographic features on each dimension for the five groups. Appendix D gives details of each demographic dimension. Note that the values for each dimension are the most prevalent ones in each group, they are not necessarily coming from a single person in that group.

