# OpenReview forum: "You Are What You Annotate: Towards Better Models through Annotator Representations"
_EMNLP/2023/Conference — EMNLP 2023 Findings_

### Official Review · Reviewer_6CyZ · 2023-08-05

**Soundness:** 2

**Excitement:**

1: Poor: I cannot identify the contributions of this paper, or I believe the claims are not sufficiently backed up by evidence. I would fight to have it rejected.

**Paper Topic And Main Contributions:**

This paper tackles an important issue in crowdsourced data, which is the handling of disagreements. They suggest that representing annotators and data instances as embeddings helps models learn better without additional computing resources.

**Questions For The Authors:**

A. How dependent is your method on what instances the annotators were assigned to label? If we were to recreate the dataset but switch up who annotates what, would you still get similar embeddings and results?

B. Besides quality, what requirements did you have for your dataset?

C. What are you trying to tell us through the TSNE plots?

**Reasons To Accept:**

The idea of representing annotators and annotations with embeddings is interesting and could potentially be used in other tasks as well.

**Reasons To Reject:**

It's not entirely clear to me what is being done here. Perhaps more organization and a diagram of the experimental setup would help.

If my understanding is correct, I'm also not convinced that the way the annotator embeddings were created is appropriate. Unless we are dealing with a very dense matrix, embeddings based on what the annotator has labeled most likely are more influenced by what the annotator has labeled and not necessarily by the annotator's preferences. For example, if an annotator is only given negative sentences to label in a sentiment analysis task, it doesn't make sense to say that that annotator has a tendency to label things as negative. (You mention this issue in the limitations section and argue that it isn't the main focus of the paper, but I argue that how you deal with missing information is crucial in your work if you're proposing a new method)

**Reproducibility:**

3: Could reproduce the results with some difficulty. The settings of parameters are underspecified or subjectively determined; the training/evaluation data are not widely available.

**Reviewer Confidence:**

3: Pretty sure, but there's a chance I missed something. Although I have a good feel for this area in general, I did not carefully check the paper's details, e.g., the math, experimental design, or novelty.

**Typos Grammar Style And Presentation Improvements:**

Make sure all the figures are referred to in the paper. For example, for Figure 3, you refer to Figure 3a, but never mention Figure 3b. In that case, do you really need that plot?

---

> ### Author Rebuttal · Authors · 2023-08-25
>
> Thank you for your feedback. As you have pointed out, “this paper tackles an important issue”. What we have done here is “representing annotators and annotations with embeddings”, and such an idea “is interesting and could potentially be used in other tasks as well”.
> - We respectfully suggest that you may not have fully understood our work. We note that **multiple other reviewers praised the clarity of our paper, as well as the broadness and concreteness of our experiments**.
>     - In terms of clarity, Reviewer 1 (Ejmf) wrote  “The paper addresses an interesting and, in my opinion, very important problem clearly” and Reviewer 2 (nSUZ) wrote, “The paper is clearly written and provides sufficient level of detail to understand the carried-out research”.
>     - In terms of the methods and experiments, Reivewer 2 (nSUZ) wrote “a novel method of integration of annotator idiosyncracies into the modeling task is presented; experiments are thorough in terms of the number of models and datasets explored, which make the findings relatively generic”. And Reviewer 3 (Yi2j) wrote, “The authors have extensively stress tested their methods across different datasets and different methods”.
> - To help with the understanding of our work, we created a [concept diagram](https://anonymous.4open.science/r/annotator-embeddings-9038/Concept-diagram.pdf) that illustrates our approach, which we plan to add to the camera-ready version of our paper.
> - For your concern: “what the annotator has labeled most likely are more influenced by what the annotator has labeled and not necessarily by the annotator's preferences.”:
>     - To make it clear, this is **not a limitation of our work**, and we **did not discuss it in the limitation section**. The reasons are as follows:
>     - Annotators are **randomly** assigned examples in the datasets we experiment on (except for those "few annotator" datasets in which each annotator may annotate the entire dataset). Therefore, the very example that the annotator annotates has nothing to do with the assignment of the rest of his examples.
>     - Let’s consider a concrete (but extreme) example. For a task with two labels, say annotator A has annotated P, P, P for the first 3 examples. For the fourth example, he may annotate P or N. He may have a tendency of P, but he very well might label it as N if the text is indeed negative. Therefore, we combine the annotation embedding together with the text embedding, with a weight parameter to balance these two. In other words, we balance the effects of annotators’ preferences versus the text.
>     - The power of capturing the annotator’s preference by using annotation embeddings lies in when the text itself is not very extreme. In those cases, the annotator’s preference may dominate how he chooses his label.
>     - The performance improvement by using annotation embeddings on various tasks and models proves the point that labels assigned by the annotator to other examples are correlated with the label on the current example. But again, **the example itself is individually independent**.
> - For Question A:
>     - Our methods work as long as the annotators are assigned examples randomly.
>     - In addition, the consistent improvements we have observed across various datasets as shown in Table 4 and Table 9, reinforce the notion that our methods can generalize well across diverse scenarios. Therefore, we anticipate similar outcomes even if we were to recreate the dataset but switch up who annotates what.
> - For Question B:
>     - In addition to quality, the other requirement is to randomly assign annotators with examples, which is how most of the datasets are collected.
> - For Question C:
>     - The TSNE plots show the existence of clusters of embeddings. For TSNE plots Figure 3 and Figure 4, there are notable clusters that emerge for annotation embedding compared to the annotator embedding. These patterns remain consistent across various hyperparameters when we run TSNE plots.
>     - Furthermore, through Figure 4 and Figure 5, we see that these emerged groups can be grounded with real-world demographic features, which underscores the practical meaning of these emerged clusters we observed in our TSNE plots.
> - For 3b, We will ensure clarity by adding a sentence that we want to compare the annotation embedding (3a) to annotator embedding (3b) in the updated version.
> - We hope to have convinced you of our contributions, and we are happy to engage in further discussion if any clarifications are still needed.

---

### Official Review · Reviewer_Yi2j · 2023-08-05

**Soundness:** 3

**Excitement:**

3: Ambivalent: It has merits (e.g., it reports state-of-the-art results, the idea is nice), but there are key weaknesses (e.g., it describes incremental work), and it can significantly benefit from another round of revision. However, I won't object to accepting it if my co-reviewers champion it.

**Missing References:**

A baseline -> https://www.jstor.org/stable/2346806
Dataset -> https://kumarde.com/papers/designing.pdf

**Paper Topic And Main Contributions:**

The authors discuss about improving new methods for modeling annotators. They have introduced those methods by adding in an extra set of layers for embedding the annotators and their annotations. The authors have field tested their methods with publicly available datasets in the domain of modeling annotations. The authors have also evaluated the methods across different methods/metrics which is useful for understanding the limits of their work.

**Questions For The Authors:**

A The method Dawid & Skyene is not considered as a baseline. Any specific reason as to why?
B The datasets SBIC/Toxic Ratings (Kumar et al.) is also not included for experiments. They consist of large amounts of data points with annotations and annotator information that can be essential.
C The datasets as a whole other than GOE is relatively small. Does this impact the generalization of your work?

**Reasons To Accept:**

1. The paper introduces an important direction in the domain of modeling annotators.
2. The authors have extensively stress tested their methods across different datasets and different methods (more in the long appendix).

**Reasons To Reject:**

The reasons to reject are in the questions and improvements. Open to hear feedback from the authors on these aspects.

**Reproducibility:**

3: Could reproduce the results with some difficulty. The settings of parameters are underspecified or subjectively determined; the training/evaluation data are not widely available.

**Reviewer Confidence:**

4: Quite sure. I tried to check the important points carefully. It's unlikely, though conceivable, that I missed something that should affect my ratings.

**Typos Grammar Style And Presentation Improvements:**

This paper is 22 pages long with all the appendix and extra results but the main 8 pages are missing key elements that is crucial for an EMNLP submission. The language aspect of it, the authors have done a good job of having lot of tables and numbers on the readers but they should think about how to include the empirical results on how their methods perform better (or not) than other baselines. The #s on tables is one thing, but a challenge in this domain is that numbers are not the full story, and individual data points do tell a different interesting story.

For sharing/uploading code for papers https://anonymous.4open.science/

---

> ### Author Rebuttal · Authors · 2023-08-25
>
> Thank you for your review, and for your appreciation of the importance of the direction we pursued and the extensive testing.
> - For Question A:
>     - Thanks for your suggestion. Upon closer examination, we found that the baseline in the suggested paper **is equivalent to the Majority Vote (MV$_{macro}$)**, one of the baseline models in our paper (with the results reported in Table 9).
>     - Conceptually, the suggested paper addresses the observation error, presupposing a single "ground truth" from which the observations may deviate. This is a different setup than the one used in our framework, where we assume that each label assigned by an annotator reflects their genuine intent.
>     - Mathematically, the difference between their settings and ours leads to that for the first step in the EM algorithm (step (2) after the initialization) from the suggested paper, equation (2.3) or (1.1) always gives us a probability of $1$ because the number of records equals to the number of the correct label in our setting. Specifically, let us take a closer look at the equation (1.1):
>         - $\hat{\pi}_{jl}^{(k)}$ = number of times observer $k$ records $l$ when $j$ is correct / number of patients seen by observer $k$ where $j$ is correct.
>         - In our setting, the observers are annotators, $l$ is the label assigned by the observer or the annotator, and $j$ in the original paper is the correct label, which in our setting is the same as $l$. Therefore, $\hat{\pi}_{jl}^{(k)}$ will always be $1$.
>         - Now consider (2.5), which is the probability of that we observe patient $i$ with label $j$, is $p(T_{ij} = 1 | data) = \Pi_{k=1}^{K}\Pi_{l=1}^{J}(\pi_{jl}^{(k)})^{n_{il}^{(k)}}p_j / \sum_{q=1}^J\Pi_{k=1}^K\Pi_{l=1}^J(\pi_{ql}^{(k)})^{n_{il}^{(k)}}p_{q}$ can be reduced to $p(T_{ij} = 1 | data) = p_j / \sum_{q=1}^Jp_q$ because the $\pi$s are $1$.
>         - $p(T_{ij} = 1 | data) = p_j / \sum_{q=1}^Jp_q = p_j$ ($j$ is the label or the observation replies) means that when the annotator or the observer labels, the probability distribution follows the overall label distribution.
>         - Therefore, if say $p_{j_1} > p_{j_2}$, we will predict label $j_1$ for that example. This **is equivalent to the Majority Vote (MV$_{macro}$) in our paper**.
>     - There are also differences between the application scenarios of the suggested paper and ours. The suggested paper pertains to domains that are subject to errors of measurement such as patient records. In contrast, our setup suits tasks that involve subjectivity, or in other words where there is no single "ground truth" label since different people have their own standards and therefore their own “ground truth” labels.
> - For Question B: the reason that we do not include the toxicity rating dataset (Dataset proposed in [Kumar et al.]( https://kumarde.com/papers/designing.pdf)) :
>     - We acknowledge the relevance of the dataset but want to clarify our reasons for its exclusion. The dataset includes annotations for each example alongside annotator demographic information. However, its data format is:
> ```json
> {
>   "comment": text,
>   "ratings": [
>       {toxic_rating_1, demographic_feature_1},
>       {toxic_rating_2, demographic_feature_2},
>       ...
>       {toxic_rating_5, demographic_feature_5}
>    ]
> }
> ```
> and **it lacks annotator identification, making it impossible to associate specific labels with corresponding annotators**. Unfortunately, this information is crucial for constructing the annotator and annotation embeddings. Therefore, our methods are incompatible with this dataset format. We nonetheless acknowledge that their work is highly relevant, and we will include this paper in the related work section in the updated version.
> - For Question C:
>     - We respectfully disagree with the statement that the datasets other than GOE (Go Emotions) are relatively small. Regarding the scale of the GOE dataset, which comprises 194k examples, it is worth noting that the other datasets also consist of a substantial number of examples, as illustrated by the following statistics (we include GOE here for a better comparison):
> | Dataset    | Number of Examples |
> | -------- | ------- |
> |GOE (Go Emotions)  |  194k |
> |HUM (Humor) | 141k |
> |SNT (Sentiment Analysis) | 60.4k |
> |MDA (MultiDomain Agreement) | 44k |
> |COM (CommitmentBank) | 11.5k |
>     - As shown in Table 4 and 9, our methods have demonstrated effectiveness across these datasets, spanning a diverse array of tasks and domains. Therefore, we are confident that our methods can generalize well.
> - Presentation improvements:
>     - Thank you for your feedback about improving the presentation. We greatly appreciate your suggestion to include additional content in the paper. In terms of the “key elements”, we ran into space issues that forced us to put content in the Appendix. As stated in our rebuttals to Reviewers 1 and 2, we will move several sections from the Appendix to the main content in the camera ready given the extra page. We will certainly enhance the comprehensive description of our findings in the updated version.
> - Lastly, here is [the anonymized repo of our project](https://anonymous.4open.science/r/annotator-embeddings-9038/). We will also make our Github repository publicly available upon acceptance.
> - We hope to have convinced you of our contributions, and we are happy to engage in further discussion if any clarifications are still needed.

---

### Official Review · Reviewer_nSUZ · 2023-08-08

**Soundness:** 3

**Excitement:**

4: Strong: This paper deepens the understanding of some phenomenon or lowers the barriers to an existing research direction.

**Paper Topic And Main Contributions:**

The paper reports on experiments focusing on the integration of annotator characteristics in modeling some subjective NLP tasks through creating annotator and annotation embeddings which are incorporated into the model. The evaluation on six NLP datasets revealed that the inclusion thereof leads to significant performance gains when learning from annotation disagreements.

**Questions For The Authors:**

- why did you put the results and the findings for the "few annotators" datasets in the annex?

- in 42-44 you mention "under-represented" groups whose opinions may not agree with the majority. I am very puzzled by this wording since a given annotator can be in one case in the majority, and in another case in the minority. So, what does this concept of "under-represented" group mean here? Wouldn't  it make sense to speak more of subjective tasks and opinions that might emerge from many factors?


**Reasons To Accept:**


- the paper is clearly written and provides sufficient level of detail to understand the carried-out research

- a novel method of integration of annotator idiosyncracies into the modelling task is presented

- experiments are thorough in terms of the number of models and datasets explored, which make the findings relatively generic

- a very thorough interpretation of the results and findings is provided


**Reasons To Reject:**


- the paper is somewhat wrongly balanced, i.e. the annex is longer than the main body of the paper and includes some relevant results, e.g., the results of the experiments on the "few annotators" datasets, which would better fit to be placed in the main body of the article


**Reproducibility:**

3: Could reproduce the results with some difficulty. The settings of parameters are underspecified or subjectively determined; the training/evaluation data are not widely available.

**Reviewer Confidence:**

5: Positive that my evaluation is correct. I read the paper very carefully and I am very familiar with related work.

**Typos Grammar Style And Presentation Improvements:**

- the datasets are referenced sometimes with full names, sometimes with the acronyms introduced earlier. Therefore, it would be better to use a consistent naming convention

- Table 4 and Table 9 are partially redundant

- maybe enumerating the main contributions in the introduction would improve the presentation

---

> ### Author Rebuttal · Authors · 2023-08-25
>
> Thank you for your review, we are glad to hear you found our experiments and the interpretation of the results to be thorough. We greatly appreciate your suggestion, and we intend to incorporate the "few annotators" results (Table 9) into the main body of our paper in the camera-ready, where an additional page is allowed. This inclusion will allow interested readers to gain a more comprehensive understanding of our methods' performance across various scenarios.
> - For Question 1:
>     - Because of the space limit, the results for "few annotator" datasets did not make it into the main text.
>     - We will include the results on "few annotator" datasets in the camera-ready version.
> - For Question 2:
>     - Thank you for engaging in a nuanced discussion about the term "under-represented".
>     - We concur that while general guidelines exist, determining hate speech is fundamentally influenced by subjective perceptions.
>     - We will change our way of framing the example and rewrite the text along the lines of: Words deemed inoffensive by white annotators may carry offensive connotations for black or Asian annotators, reflecting the cultural intricacies that contribute to the subjective nature of hate speech detection.
> - For presentation improvement:
>     - Thanks for your suggestions! In the updated version of our paper, we will make sure to be consistent in the naming of the dataset, we will move Table 9 to the main text and drop Table 4, and we will add a paragraph in the introduction section to specifically list  our contributions.

---

### Official Review · Reviewer_Ejmf · 2023-08-09

**Soundness:** 3

**Excitement:**

4: Strong: This paper deepens the understanding of some phenomenon or lowers the barriers to an existing research direction.

**Paper Topic And Main Contributions:**

While the general practice is to force annotators’ agreement through aggregation strategies (e.g., majority voting), many tasks are legitimately subjective, so the disagreement among annotators can be due to their different points of view rather than, e.g. noise or bad guidelines.

This paper proposes a method to leverage and study annotators’ disagreement by embedding annotators and their annotations.

Experiments are performed on a large set of diverse datasets, and the results are analyzed from a wide variety of points of view.


**Questions For The Authors:**

Question A: How are unknown annotators (e.g., last paragraph of section 8) modeled? Is a zeroed embedding vector produced? Is there an embedding for all unknown annotators?

Question B: In section 3, the problem is framed as that of maximizing the annotator-specific correct label. Am I correct that this is how the models are evaluated in Section 6?

Question C: While I understand the paper's focus is on the analyses rather than on improving performance per se (a focus that I appreciate), I think the experiment considering unknown annotators should be given more relevance. This is because the paper still emphasizes “practical” aspects, e.g., performance improvement and model efficiency. While it is nice to show that the performance improves when knowing the specific annotator’s ID, in the majority of practical use cases, the set of annotators of the training set and those of the model target “users” are disjoint; however, it might be possible to collect the user’s opinion on a subset of instances, similarly to a cold start scenario in a recommender system. I would give more relevance to this experiment, for which results are currently relegated to the Appendix. I would also be interested in knowing the relationship between the number of available annotations for unknown annotators and the performance.

**Reasons To Accept:**

The paper addresses an interesting and, in my opinion, very important problem clearly.

It has two main focuses: on the one hand, it shows that performance improves when adding annotators and annotations embedding to a simple transformer model (interestingly, the method increases the number of parameters by 1% only).

On the other hand, the paper performs a deep analysis of what the embeddings convey, their contributions to the performance, and many other interesting aspects.

Many different datasets are explored (different tasks, number of annotators, number of instances, etc). References to previous work are clear and frame the problem well.

**Reasons To Reject:**

In my opinion, the paper's contribution comes more from the performed analysis than from the increase in performance per se.

This is because the paper assumes a closed set of annotators for which classification will be performed. This is far from most practical scenarios, where models are trained once and then used by different target users.

**Reproducibility:**

4: Could mostly reproduce the results, but there may be some variation because of sample variance or minor variations in their interpretation of the protocol or method.

**Reviewer Confidence:**

3: Pretty sure, but there's a chance I missed something. Although I have a good feel for this area in general, I did not carefully check the paper's details, e.g., the math, experimental design, or novelty.

**Typos Grammar Style And Presentation Improvements:**

I think that adding a picture depicting the architecture and how the embedding and the matrices described in section 4 fit in it would make the understanding of the sections much more intuitive.

---

> ### Author Rebuttal · Authors · 2023-08-25
>
> Thank you for your feedback! We are grateful to see that you recognized the significance of the analysis in our work.
>
> - For Rejection Reason 2 & Question A:
>     - From an application perspective, the performance gains for known annotators show that if we have accumulated a certain amount of data on certain people, we can leverage those to gain insights into individual preferences and accommodate them effectively.
>     - For unknown annotators, we predict their labels by using an annotator embedding that is randomly initialized like the rest of the model parameters, without any tuning. As there is no corresponding annotated data in the training set for the unknown annotators, there will be zero annotation embeddings for these annotators.
>     - We acknowledge that future efforts are needed and we specifically consider pursuing a direction where we derive the embeddings for unknown annotators from embeddings of known annotators who exhibit similar demographic features, especially given the correlations between the embeddings and demographic features discussed in Section 8.
> - For Question B:
>     - You are correct, the metrics, namely exact match accuracy and macro F1 scores, are measured on the annotator-specific correct label.
> - For Question C:
>     - Thank you for your suggestion! We ran into space constraints for the submitted version, but we will consider moving the relevant part from the Appendix to the main body of the paper for the camera-ready version.
> - For presentation improvements:
>     - Thanks for your suggestion, and we are considering adding [this diagram](https://anonymous.4open.science/r/annotator-embeddings-9038/Concept-diagram.pdf) to our paper.

---

### Meta-Review · Area_Chair_resA · 2023-09-09

**Recommendation:** 4

**Metareview:**

This paper considers the problem of learning subjective tasks where annotators might have different perspectives, and proposes to capture those perspectives with "annotator embeddings." Results show that these embeddings improve performance, and the paper additionally carries out a large number of analyses to understand the learning process.

The main strength of this paper is the analysis of the approach, which is incredibly detailed and very interesting to read. I suspect that the analysis will be the thing remembered about this paper.

The main weakness of this paper is the setting - it assumes that the training and test data contain the same annotators. As a result, this almost reads more as a "personalization" paper than an "annotator modeling" paper. Reading it under that view does not - to me - detract from the paper, and in some ways makes it more interesting. The paper does not make a case for why the "same annotator" setting is realistic/interesting.

The "unknown annotators" experiment somewhat addresses this question (in the personalization language, this is the "cold start problem"). However, as AC, I do not understand this result. Based on the paper and the discussion with reviewer Ejmf, my understanding is that none of the reviewers at test time are seen at training time, that they're given a random embedding, and yet, even in that situation, there is no large degradation in performance from the Text Only baseline. But all this says is that having random embeddings doesn't hurt over having no embeddings, which is not saying much. The more interesting comparison is to the results from Table 4. As an attempt to reconstruct them (comparing to E_n+E_a for instance), we have:

            E_n+E_a                E_n+E_a
            (known annots)      (unnkown annots)
  MDA        75.76                      74.24
  HUM        53.89                      53.51
  COM        44.41                      40.28
  GOE        69.90                      61.96
  SNT         64.61                      37.90

Here, MDA and HUM show basically no difference, COM shows a reasonably big difference (4%) and GOE and SNT show large differences (8% and 27% respectively). However, in Table 4, MDA and HUM are the ones where E_n+E_a doesn't really improve over Text Only, and GOE and SNT are the ones where they do.

Basically all this seems to be saying is that when this approach provides big gains (in Table 4), it no longer does (in Table 6).

This is not at all surprising - there's no reason to even expect that this would work.

This is all to say that I agree with this weakness - though thinking of this as personalization rather than annotator disagreement mitigates this - and I don't think the experiment described really attenuates the concern.

---

### Decision · Program_Chairs · 2023-10-07

**Decision:**

Accept-Findings

**Comment:**

This paper considers the problem of learning subjective tasks where annotators might have different perspectives, and proposes to capture those perspectives with "annotator embeddings." Results show that these embeddings improve performance, and the paper additionally carries out a large number of analyses to understand the learning process.

The main strength of this paper is the analysis of the approach, which is incredibly detailed and very interesting to read. I suspect that the analysis will be the thing remembered about this paper.

The main weakness of this paper is the setting - it assumes that the training and test data contain the same annotators. As a result, this almost reads more as a "personalization" paper than an "annotator modeling" paper. Reading it under that view does not - to me - detract from the paper, and in some ways makes it more interesting. The paper does not make a case for why the "same annotator" setting is realistic/interesting.

The "unknown annotators" experiment somewhat addresses this question (in the personalization language, this is the "cold start problem"). However, as AC, I do not understand this result. Based on the paper and the discussion with reviewer Ejmf, my understanding is that none of the reviewers at test time are seen at training time, that they're given a random embedding, and yet, even in that situation, there is no large degradation in performance from the Text Only baseline. But all this says is that having random embeddings doesn't hurt over having no embeddings, which is not saying much. The more interesting comparison is to the results from Table 4. As an attempt to reconstruct them (comparing to E_n+E_a for instance), we have:

            E_n+E_a                E_n+E_a
            (known annots)      (unnkown annots)
  MDA        75.76                      74.24
  HUM        53.89                      53.51
  COM        44.41                      40.28
  GOE        69.90                      61.96
  SNT         64.61                      37.90

Here, MDA and HUM show basically no difference, COM shows a reasonably big difference (4%) and GOE and SNT show large differences (8% and 27% respectively). However, in Table 4, MDA and HUM are the ones where E_n+E_a doesn't really improve over Text Only, and GOE and SNT are the ones where they do.

Basically all this seems to be saying is that when this approach provides big gains (in Table 4), it no longer does (in Table 6).

This is not at all surprising - there's no reason to even expect that this would work.

This is all to say that I agree with this weakness - though thinking of this as personalization rather than annotator disagreement mitigates this - and I don't think the experiment described really attenuates the concern.